# Aligning Multimodal Representations through an Information Bottleneck

**Antonio Almudévar** [1]  **José Miguel Hernández-Lobato** [2]  **Sameer Khurana** [3]  **Ricard Marxer** [4]  **Alfonso Ortega** [1]

## Abstract

Contrastive losses have been extensively used as a tool for multimodal representation learning. However, it has been empirically observed that their use is not effective to learn an aligned representation space. In this paper, we argue that this phenomenon is caused by the presence of modality-specific information in the representation space. Although some of the most widely used contrastive losses maximize the mutual information between representations of both modalities, they are not designed to remove the modality-specific information. We give a theoretical description of this problem through the lens of the Information Bottleneck Principle. We also empirically analyze how different hyperparameters affect the emergence of this phenomenon in a controlled experimental setup. Finally, we propose a regularization term in the loss function that is derived by means of a variational approximation and aims to increase the representational alignment. We analyze in a set of controlled experiments and real-world applications the advantages of including this regularization term.

## 1. Introduction

Multimodal Learning is an area of AI that is focused on processing and integrating information from multiple modalities (e.g., text, image or audio). It is becoming a pivotal topic in the community because of multiple reasons, including, but not limited to, (i) it permits to mimic human cognition processes (Fei et al., 2022; Lee et al., 2023); (ii) it allows to use a greater amount of training data from different modalities, which tends to improve the performance of the models

(Kaplan et al., 2020; Cuervo & Marxer, 2024); and (iii) it is essential in real-world applications such as autonomous vehicles (Xiao et al., 2020), healthcare (Kline et al., 2022) or human-computer interaction (Sinha et al., 2010).

Similarly to humans, most AI systems work through obtaining intermediate representations, which are compressed versions of the raw data that preserve useful information to solve different downstream tasks (Bengio et al., 2013; Cadieu et al., 2014). One of the most widely used ways of training multimodal systems is Contrastive Representation Learning (Karpathy & Fei-Fei, 2015; Oord et al., 2018; Tian et al., 2020a). In this paradigm, representations corresponding to similar input data are brought closer than dissimilar ones. For example, the caption "a photo of a dog" should become closer to an image of *a dog* than to that of *a cat*. The most widely used of the contrastive losses is the InfoNCE (Oord et al., 2018). Minimizing this loss is equivalent to maximizing a lower bound of the mutual information of the representations from both modalities. In other words, when minimizing this loss, representations from each modality should maximize the information that they contain about what is common between them.

However, the above does not imply that representations from both modalities contain the same information. Representations could contain all the information about what is common to both modalities, but still preserve much of the information that is specific to their own modality (a.k.a. nuisances from now on). We argue that this can translate into a substantial representational misalignment (Klabunde et al., 2023), especially when the inputs contain a high level of nuisances. In other words, representations from two modalities of a positive pair could be not so similar to each other due to the fact that they are encoding different information. Figure 1 illustrates a trivial example in which two similar, yet different, images have exactly the same caption. Thus, the text representations are exactly the same, while the image representations are different from each other, since they can be encoding information about aspects like the color of the dog, the number of clouds in the sky or the number of blades of grass. This misalignment phenomenon has been already observed and denominated *modality gap* (Liang et al., 2022). However, to the best of our knowledge, the present is the first work in which this phenomenon is explained from an information theory perspective.

[1]ViVoLab, Aragón Institute for Engineering Research (I3A), University of Zaragoza, Zaragoza, Spain [2]University of Cambridge, Cambridge, UK [3]Mitsubishi Electric Research Laboratories (MERL), Cambridge, USA [4]Université de Toulon, Aix Marseille Univ, CNRS, LIS, Toulon, France. Correspondence to: Antonio Almudévar <almudevar@unizar.es>.

*Proceedings of the 42$^{nd}$ International Conference on Machine Learning*, Vancouver, Canada. PMLR 267, 2025. Copyright 2025 by the author(s).

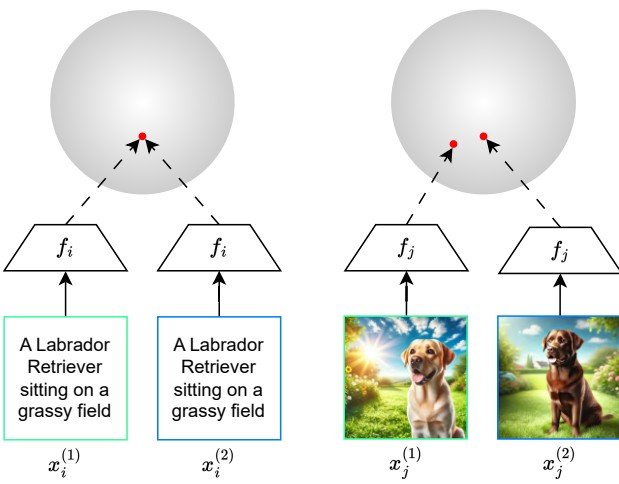

Figure 1: Different modalities usually contain different information. A trivial example of this is the case in which two different images have exactly the same caption. As a consequence of this, representations from different modalities tend to contain different information (thus leading to misalignment) if the opposite is not explicitly imposed.

It is precisely this explanation which allows us to propose a solution to this phenomenon. Concretely, we propose to apply an Information Bottleneck (IB) (Tishby et al., 2000) in the representation space. With this, apart from maximizing the mutual information between the representations of both modalities through the contrastive loss, we reduce the nuisances that can be found in the representations. This IB is applied by means of a regularization term in the loss function that is derived using a variational approximation. Thus, it is efficient, straightforward to implement and modality-agnostic, which is advantageous over alternative approaches (Li et al., 2021).

## 2. Preliminaries

**Contrastive Representation Learning (CRL)** This encompasses a set of techniques that learns a representation space in which representations of similar inputs are closer than those of dissimilar ones. It has emerged as one of the most competitive methods for learning representations without labels in a self-supervised way (Oord et al., 2018; Hjelm et al., 2018; Wu et al., 2018; Logeswaran & Lee, 2018; Bachman et al., 2019; Tian et al., 2020a; Chen et al., 2020a; Henaff, 2020). The most widely used among the contrastive losses is the InfoNCE (Oord et al., 2018) and it has been shown that minimizing this is equivalent to maximizing a lower bound of the mutual information (MI) between a pair of representations (Bachman et al., 2019; Tian et al., 2020a).

**Multimodal Contrastive Representation Learning** One of the tasks in which contrastive losses have gained popularity is Multimodal Representation Learning, which consists in designing systems that map inputs from different

modalities (e.g. image and text) into a joint representation space. Some foundation works used rank-based losses (Yager, 1988; Usunier et al., 2009; Schroff et al., 2015) to learn multimodal representation spaces (Weston et al., 2010; Frome et al., 2013; Karpathy & Fei-Fei, 2015) while more modern approaches have used the InfoNCE loss (Tian et al., 2020a; Radford et al., 2021; Jia et al., 2021; Xu et al., 2021; Girdhar et al., 2023) for this purpose. However, it has been observed that, when trained in a contrastive way, representations from different modalities tend to be located in different regions of the space, a phenomenon called *modality gap* (Liang et al., 2022; Udandarao, 2022; Ramasinghe et al., 2024; Fahim et al., 2024; Schrodi et al., 2024). This phenomenon can be an issue in some applications such as Image Captioning or Visual Question Answering, so sophisticated training methods have been proposed to palliate it (Chen et al., 2020b; Li et al., 2021; 2022; 2023).

**Measuring Representational Alignment** The representational alignment (or similarity) is typically measured through kernel alignment metrics (Cristianini et al., 2001; Cortes et al., 2012). Examples of these include Centered Kernel Alignment (CKA) (Kornblith et al., 2019), SVCCA (Raghu et al., 2017) and nearest-neighbor metrics (Klabunde et al., 2023). However, in this work we restrict our attention to the former, since it is the most widely used for this purpose. Let $Z^{(\alpha)} \in \mathbb{R}^{n \times d_\alpha}$ and $Z^{(\beta)} \in \mathbb{R}^{n \times d_\beta}$ be two sets of representations, $K = k(z_i^{(\alpha)}, z_j^{(\alpha)})$ and $L = l(z_i^{(\beta)}, z_j^{(\beta)})$, where $k : \mathbb{R}^{d_\alpha} \times \mathbb{R}^{d_\alpha} \to \mathbb{R}$ and $l : \mathbb{R}^{d_\beta} \times \mathbb{R}^{d_\beta} \to \mathbb{R}$ are kernel functions. Then, the Hilbert-Schmidt Independence Criterion is defined as:

$$HSIC(K, L) = \frac{1}{(n-1)^2} \operatorname{Tr}(KHLH) \qquad (1)$$

where $H = I_n - \frac{1}{n}\mathbf{1}\mathbf{1}^T$ is the centering matrix. Then, the CKA is defined as follows:

$$CKA(K, L) = \frac{HSIC(K, L)}{\sqrt{HSIC(K, K)HSIC(L, L)}} \qquad (2)$$

This metric ranges between zero and one and we say that a pair of representations is perfectly aligned when $CKA = 1$.

**Information Bottleneck** It is a framework that aims to find a representation that contains all the information in an input that is necessary to solve a given task, while discarding irrelevant information (Tishby et al., 2000; Tishby & Zaslavsky, 2015). Given an input $X$, a task $Y$ and a representation $Z$, the goal can be formulated as:

$$\max_Z I(Z; Y) - \beta I(Z; X) \qquad (3)$$

where $\beta$ is a Lagrange multiplier that controls the trade-off between compression and preserving the information that is relevant for the task $Y$.

# 3. On the Importance of Minimal Sufficient Representations

To formulate our hypothesis, we assume that the data of a modality are composed of an essence, which is all that information that can be found in both modalities; and nuisances, which refers to all that information that can be found only in one modality. In addition, our goal is to obtain representations of the inputs of each modality. Next, we explain more in detail these concepts. All the proofs of the Lemmas and Theorems can be found in Appendix A

## 3.1. Input Data: Essence and Nuisances

**Definition 1.** Let $(X_\alpha, X_\beta)$ be a pair of positive inputs from modalities $\alpha$ and $\beta$, respectively. We call *essence* to a variable $Y$ that satisfies the following Markov Chains:

$$X_\beta \leftrightarrow Y \leftrightarrow X_\alpha \qquad (4)$$
$$Y \leftrightarrow X_\beta \leftrightarrow X_\alpha \qquad (5)$$

i.e., it refers to the common part of a positive pair of data. Although there exists more than one essence (infinite, in fact), all of them are equivalent under a one-to-one transformation, which is formalized next.

**Lemma 1.** *Let $Y$ and $Y'$ be essences of the same pair of modalities. Then, there exist a one-to-one transformation $\Psi$ such that $Y = \Psi(Y')$.*

Equivalently, the partitions of $X_\alpha$ and $X_\beta$ created by $Y$ are unique. We note that the essence $Y$ is a variable that we define to help us with the formulation, but our goal is not to discover $Y$ itself, but the partition of the input set that it creates. For example, if we have two images with the same caption, we will consider that both images are equivalent in the sense that they belong to a common subset of the images set, but we are not interested in defining the subset. Thus, from now on we will refer to the essence as if it were a unique variable.

**Definition 2.** Let $X_\alpha$ be an input of modality $\alpha$, $Y$ the essence of $X_\alpha$ with respect to another modality. We call *nuisance* of modality $\alpha$ to a variable $N_\alpha$ that satisfies:

$$I(Y; N_\alpha) = 0 \qquad (6)$$
$$I(X_\alpha; N_\alpha) = H(X_\alpha) - H(Y) = H(N_\alpha) \qquad (7)$$

i.e., it refers to all information from $X_\alpha$ that cannot be found in the other modality.

Diagram in Figure 2 schematizes the relationships between all the elements described in this section.

## 3.2. Representations

**Definition 3.** We say that a variable $Z_\alpha$ is a *representation* of an input $X_\alpha$ if $Z_\alpha$ is a stochastic function of $X_\alpha$ or, equivalently, if $Z_\alpha$ is fully defined by $p(z_\alpha|x_\alpha)$.

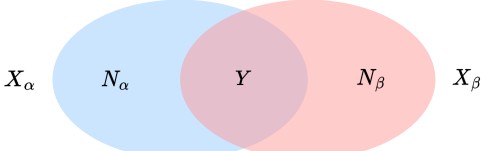

Figure 2: Diagram of the inputs, essence and nuisances

Given a representation $Z_\alpha$ of $X_\alpha$, the following Markov chains are satisfied:

$$Y \leftrightarrow X_\alpha \leftrightarrow Z_\alpha \qquad (8)$$
$$N_\alpha \leftrightarrow X_\alpha \leftrightarrow Z_\alpha \qquad (9)$$

The goal of representation learning is to obtain representations with desirable properties for the problem to be solved. In the case of multimodal learning we consider two properties to be desirable: (i) sufficiency and (ii) minimality (Achille & Soatto, 2018a;b). We define these properties and argue their desirability next.

**Definition 4.** Given a representation $Z_\alpha$ of the input $X_\alpha$ and an essence $Y$. We call $Z_\alpha$ *sufficient* if it satisfies:

$$I(Z_\alpha; Y) = I(X_\alpha; Y) \qquad (10)$$

i.e., a representation is sufficient if it preserves the essence in its entirety or, equivalently, if it preserves all the information that is common to both modalities. Because of Equation (8) we know by the Data Processing Inequality (DPI) that $I(Z_\alpha; Y) \leq I(X_\alpha; Y)$, i.e. $I(X_\alpha; Y)$ is an upper bound of $I(Z_\alpha; Y)$. Thus, the objective to be optimized to obtain a sufficient representation is:

$$\max_{Z_\alpha} I(Z_\alpha; Y) \qquad (11)$$

Informally, *sufficiency is connected to the performance of our representations in downstream tasks*. Our representations must have all the information in the essence to perfectly solve *all* the tasks that can be derived from the essence. Here, we assume that tasks that are not in the essence cannot be solved. For example, if we have a set of images of dogs and a set of captions describing different aspects of them except the color, we cannot expect the text encoder to be able to *understand* the word "brown" even if there are brown dogs in our set of images. We formalize this next.

**Theorem 1.** *Let $Y$ and $Z_\alpha$ be the essence and a representation of input $X_\alpha$ respectively, and let $\mathcal{T} = \{T : T = f(Y)\}$ be the set of all the deterministic functions of $Y$ (i.e., all the tasks derived from $Y$). Then, we have that:*

$$p(t|z_\alpha) = p(t|x_\alpha) \; \forall \, T \in \mathcal{T} \implies I(Z_\alpha; Y) = I(X_\alpha; Y)$$

**Definition 5.** Given a representation $Z_\alpha$ and the nuisances $N_\alpha$ of an input $X_\alpha$. We call $Z_\alpha$ *minimal* if it satisfies:

$$I(Z_\alpha; N_\alpha) = 0 \qquad (12)$$

i.e., a representation is minimal if it eliminates all the nuisances of its modality or, in other words, if all the information that it contains can also be found in the input of the other modality. Since mutual information is non-negative, the objective to be optimized in order to obtain a minimal representation is:

$$\min_{Z_\alpha} I(Z_\alpha; N_\alpha) \tag{13}$$

Informally, *minimality is connected to representational alignment*. As explained in section 2, we can have representations with a good performance in a wide variety of downstream tasks but misaligned. Intuitively, even if two representations $Z_\alpha$ and $Z_\beta$ have all the information about the essence (i.e., they are sufficient), if they also contain information about nuisances (i.e., they are not minimal), then the information they are encoding is different and, consequently, they will be misaligned. Revisiting the previous example, the sufficient representations of an image of *a **yellow** Labrador Retriever* and of an image of *a **brown** Labrador Retriever* could be different from each other, since they can be coding different information. However, the sufficient representations corresponding to the captions of each image will be presumably the same, since both images have presumably the same caption. Therefore, the presence of nuisances in the representations could cause misalignment, as stated next.

**Theorem 2** (Informal). *Let $Z_\alpha$ and $Z_\beta$ be the representation of some inputs with nuisances $N_\alpha$ and $N_\beta$, respectively, such that $Z_\alpha$ and $Z_\beta$ are aligned in the sense of equation (2). Then, $I(Z_\alpha; N_\alpha) = I(Z_\beta; N_\beta) = 0$.*

Summarizing the above, we have that our ideal representation should be a minimal sufficient statistic for $Y$. Equivalently, it should contain *only* (minimal) *all* (sufficient) the information that is common to both modalities (a.k.a. the essence). In this scenario, similarly to Lemma 1, we know that all the ideal representations create the same partition of the input. This connects to the "Anna Karenina principle" [1] that has been mentioned in different works in representation learning to hypothesize that all the well-performing models learn roughly the same internal representations (Bansal et al., 2021; Huh et al., 2024). Here, all the minimal sufficient ("*happy*") representations ("*families*") create the same partition of the input ("*are alike*").

## 4. Obtaining Minimal Sufficient Representations

We have discussed in the previous section the importance of minimal sufficient representations for good performance

---

[1] "All happy families are alike; each unhappy family is unhappy in its own way." (Tolstoy, 1877). This principle was popularized in (Diamond & Ordunio, 1999) to illustrate why only a small number of wild animals have been successfully domesticated over the course of history.

and alignment. We describe next a method to obtain them, which connects to the Information Bottleneck principle.

### 4.1. Obtaining Sufficient Representations

Equation (11) shows that $I(Z_\alpha; Y)$ must be maximized to find a representation $Z_\alpha$ that is sufficient. Since the essence $Y$ is a variable that we have defined for our formulation whose distribution is unknown, calculating this term could seem problematic. However, in Appendix A.4 we show that $I(Z_\alpha; Y) = I(Z_\alpha; X_\beta)$. Thus, our objective becomes

$$\max_{Z_\alpha} I(Z_\alpha; X_\beta) \tag{14}$$

Computing exactly $I(Z_\alpha; X_\beta)$ is in general intractable, since it involves integrating over the entire space of $\beta$-modality inputs. However, we can obtain a lower bound of $I(Z_\alpha; X_\beta)$: since $Z_\beta$ is a representation of $X_\beta$, we have that $Z_\alpha \leftrightarrow X_\beta \leftrightarrow Z_\beta$ and, by the DPI, it follows that $I(Z_\alpha; Z_\beta) \leq I(Z_\alpha; X_\beta)$. That is, $I(Z_\alpha; Z_\beta)$ is a lower bound of $I(Z_\alpha; Y)$. Analogously, $I(Z_\beta; Z_\alpha) \leq I(Z_\beta; Y)$, so given the symmetry of the mutual information, we must maximize $I(Z_\alpha; Z_\beta)$ in order to jointly maximize $I(Z_\alpha; Y)$ and $I(Z_\beta; Y)$. Thus, the objective becomes:

$$\max_{Z_\alpha, Z_\beta} I(Z_\alpha; Z_\beta) \tag{15}$$

Again, computing exactly $I(Z_\alpha; Z_\beta)$ requires integrating over the representation spaces, which is in general intractable. However, as explained in section 2, minimizing InfoNCE loss is equivalent to maximizing a lower bound of $I(Z_\alpha; Z_\beta)$. Thus, *encoders optimized through* InfoNCE *tend to give sufficient representations*. However, the resulting representations are not necessarily minimal due to the fact that this loss imposes no conditions on $I(Z_\alpha; N_\alpha)$. We derive in the next section a term that aims to increase the degree of minimality of the representations.

### 4.2. Obtaining Minimal Representations

Equation (13) shows that $I(Z_\alpha; N_\alpha)$ must be minimized to obtain a representation $Z_\alpha$ that is minimal. Since $N_\alpha$ is an abstract concept whose distribution is unknown, calculating this term could seem problematic. However, by the DPI and equation (9), we have that $I(Z_\alpha; N_\alpha) \leq I(Z_\alpha; X_\alpha)$. Thus, the objective to obtain a minimal representation becomes:

$$\min_{Z_\alpha} I(Z_\alpha; X_\alpha) \tag{16}$$

Again, computing exactly $I(Z_\alpha; X_\alpha)$ requires integrating over the representation and input spaces, which is intractable. However, we demonstrate in Appendix A.5 that:

$$I(Z_\alpha; X_\alpha) \leq \mathbb{E}_{p(x_\alpha, x_\beta)} \left[ D_{KL} \left( p_{\theta_\alpha}(z|x_\alpha) || p_{\theta_\beta}(z|x_\beta) \right) \right] \tag{17}$$

Therefore, we can minimize the given upper bound to minimize $I(Z_\alpha; X_\alpha)$. That is, the distributions of the representations of a positive pair of data from different modalities should be as similar as possible. Intuitively, if $Z_\alpha$ and $Z_\beta$ are equal, then they can be affected only by the essence but not by the nuisances.

**Spherical Gaussian Case**   The upper bound of equation (17) does not have a closed form in general. However, it is common to assume in practice that the representations distributions given the input are Gaussian, in which case, KL Divergence becomes tractable. In the case in which $p_{\theta_\alpha}(z|x_\alpha) = \mathcal{N}\left(z; \mu_{\theta_\alpha}(x_\alpha), \sigma^2 I\right)$ and $p_{\theta_\beta}(z|x_\beta) = \mathcal{N}\left(z; \mu_{\theta_\beta}(x_\beta), \sigma^2 I\right)$, as shown in Appendix A.6, we reach:

$$
\begin{aligned}
\mathop{\mathbb{E}}_{p(x_\alpha, x_\beta)} & \left[ D_{KL}\left(p_{\theta_\alpha}(z|x_\alpha) \| p_{\theta_\beta}(z|x_\beta)\right)\right] \propto \\
& \mathop{\mathbb{E}}_{p(x_\alpha, x_\beta)}\left[\left\|\mu_{\theta_\alpha}(x_\alpha) - \mu_{\theta_\beta}(x_\beta)\right\|_2^2\right] = \mathcal{L}_M
\end{aligned}
\tag{18}
$$

### 4.3. Information Bottleneck for two Modalities

Combining equations (14) and (16), the objective to obtain a representation $Z_\alpha$ that is sufficient and minimal becomes:

$$
\max_{Z_\alpha} I(Z_\alpha; X_\beta) - \beta I(Z_\alpha; X_\alpha)
\tag{19}
$$

This is equivalent to an information bottleneck in which the task is the input of the other modality $X_\beta$. That is, $Z_\alpha$ must retain *only all* the information that is common between $X_\alpha$ and $X_\beta$. The same applies for $Z_\beta$. Combining equations (15) and (18), we have that this is equivalent to minimizing:

$$
\mathcal{L} = \mathcal{L}_{\text{InfoNCE}} + \beta \mathcal{L}_M
\tag{20}
$$

## 5. Toy Experiment

The objectives of this experiment are to empirically validate the different statements made throughout the previous sections and understand the relations between the different elements of our formulation. For this purpose, we use some datasets typically employed in disentanglement related tasks (Wang et al., 2024). Concretely, DSprites (Matthey et al., 2017), MPI3D (Gondal et al., 2019) and Shapes3D (Burgess & Kim, 2018) are used. These datasets contain images and labels that represent multiple independent factors of variation. We jointly train an image encoder and a factors encoder (i.e., images and factors are the two modalities). The reason to use these datasets is that we can control the amount of factors that we input to the encoder, thus controlling the information imbalance between both modalities. Unless otherwise stated, a ResNet20 (He et al., 2016) is used as image encoder, an MLP as encoder for the factors [2] and temperature in the InfoNCE loss is a trainable parameter initialized to 0.07. More details are given in Appendix C.

---

[2]We encode the factors using one-hot.

### 5.1. Does the contrastive loss alone remove nuisances?

To answer this question we propose several scenarios per dataset. In each scenario we provide all but one factor to the encoder, i.e., the nuisances of the image modality $N_\alpha$ are that missing factor. Thus, if the contrastive loss were eliminating the nuisances, then the image representations $Z_\alpha$ should contain no information about $N_\alpha$, i.e., $I(Z_\alpha; N_\alpha) = 0$. We calculate a lower bound of this mutual information $\hat{I}(Z_\alpha, N_\alpha)$ by training a linear classifier from $Z_\alpha$ to $N_\alpha$, following (Xu et al., 2020). We show in Table 1 the values of $\frac{\hat{I}(Z_\alpha; N_\alpha)}{H(N_\alpha)}$ for each dataset and category of factors[3]. This value encodes a lower bound of the ratio of the uncertainty of $N_\alpha$ that is reduced by observing $Z_\alpha$ and its value ranges from 0 to 1 (we call it uncertainty reduction ratio or simply URR), so if the contrastive loss alone removed all the nuisances, its value would be zero. We can extract the following conclusions from this: (i) a non-negligible amount of information about the missing factors is present in the image representation for every category; (ii) image encoder preserves more information on some categories than on others; and (iii) some categories are almost equally conserved among the datasets. We hypothesize that the last two points could be due to the inductive biases of the convolutional architecture of the image encoder (Cohen & Shashua, 2016; Mitchell, 2017; Wang & Wu, 2023), but exploring this point is out of the scope of this work.

Table 1: URR (in percentages) for each dataset and category. Some factors are not used because they do not fall into any category.

|  | DSprites | MPI3D | Shapes3D |
|---|---|---|---|
| Location | $16.1 \pm 3.7$ | $12.4 \pm 7.3$ | $8.5 \pm 0.1$ |
| Shape | $77.1 \pm 4.6$ | $10.3 \pm 1.0$ | $8.7 \pm 0.4$ |
| Size | $66.3 \pm 3.1$ | $37.8 \pm 0.9$ | $7.2 \pm 0.5$ |
| Objects Color | $-$ | $68.8 \pm 2.3$ | $54.1 \pm 2.0$ |

**Not all architectures remove nuisances to the same extent** It is well established that different neural architectures introduce distinct inductive biases (Raghu et al., 2021). Consequently, the extent to which nuisance factors are retained in the learned representations can vary depending on the model architecture. To investigate this, we replicate the previous experiment using a small Vision Transformer (ViT) (Dosovitskiy et al., 2020) as the image encoder. Table 2 reveals two key observations: (i) local information—particularly *Location*—is less preserved in ViTs, likely due to their global attention mechanisms favoring long-range dependencies; and (ii) more global features—such as *Color*—are comparably preserved in both convolutional and transformer-based models. We emphasize that these trends may also depend

---

[3]We organize the factors into categories for ease of understanding of the conclusions. Information on what factors each category is composed of is provided in Appendix C.

on other architectural choices, such as the encoder depth, as explored in the following paragraph.

Table 2: URR (in percentages) for each dataset and category for ViT-based encoder.

|               | DSprites      | MPI3D         | Shapes3D      |
|---------------|---------------|---------------|---------------|
| Location      | $2.8 \pm 0.6$ | $2.8 \pm 0.3$ | $1.1 \pm 0.1$ |
| Shape         | $64.9 \pm 1.4$| $7.0 \pm 0.4$ | $8.7 \pm 0.2$ |
| Size          | $30.7 \pm 3.5$| $20.8 \pm 3.5$| $6.9 \pm 1.5$ |
| Objects Color | –             | $63.5 \pm 9.8$| $53.5 \pm 1.6$|

**Deeper neural encoders remove more nuisances** It has been argued that the success of Deep Learning can be explained through the fact that deep neural networks implicitly introduce an Information Bottleneck (Tishby & Zaslavsky, 2015; Shwartz-Ziv & Tishby, 2017). Intuitively, deterministic layers tend to remove information from the input because of the DPI. Thus, when the number of layers grows, the output of the neural network tends to be a more pruned version of the input but that preserves the information that is necessary to solve different downstream tasks (Alemi et al., 2016). We hypothesize then that the use of deeper neural encoders will tend to remove more nuisances. Effectively, we can observe in Figure 3 a trend among different factors in which deeper encoders remove more modality specific information. This can serve as an explanation for *The Capacity Hypothesis* stated in (Huh et al., 2024), which says that bigger models are more likely to converge to a shared representation than smaller models. We hypothesize that this representation is shared because it contains little information about the nuisances.

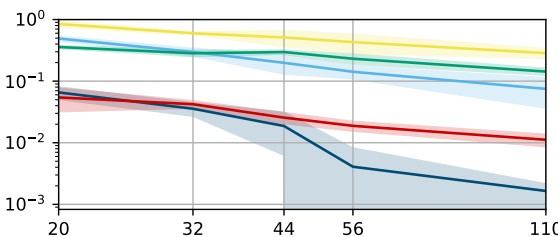

Figure 3: URR (y-axis) for different number of layers of the image neural encoder (x-axis). Same legend as Figure 4.

**Higher temperatures remove more nuisances** Wang & Liu (2021) observed that the value of the temperature in the InfoNCE loss considerably impacts on the entropy level of the representations. As stated in section 3.2, alignment is closely related to the level of nuisances in the representations and, consequently, to their entropy. We run an experiment identical to the previous one for some factors in which the temperature is fixed. Its results are shown in Figure 4 and we observe that (i) higher values of temperature tend to remove more nuisances and (ii) not all the factors are equally affected by the changes in temperature.

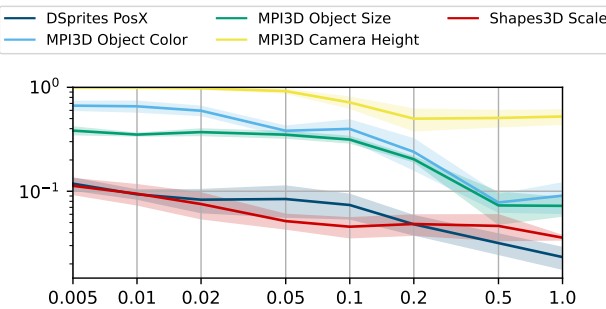

Figure 4: URR (y-axis) for different values of temperature (x-axis).

### 5.2. Does the presence of nuisances in the representation negatively correlate with the level of alignment?

As informally demonstrated in section 3.2, the fact that two representations are minimal is a necessary condition for them to be aligned. We hypothesize that misalignment is just an effect of an information imbalance in the representation space. To empirically analyze this phenomenon, we design an experiment similar to the previous one. In this case, more than one factor can be removed, i.e., $N_\alpha$ can be a set of factors. We generate 100 scenarios per dataset in which a randomly chosen subset of factors $N_\alpha$ is not provided as input to the factors encoder. Similarly to the previous experiment, we calculate $\hat{I}(Z_\alpha; N_\alpha)$ and the $CKA$ metric. In Figure 5 it is shown that, for the three datasets, there exists a negative correlation between $\hat{I}(Z_\alpha; N_\alpha)$ and the alignment value.

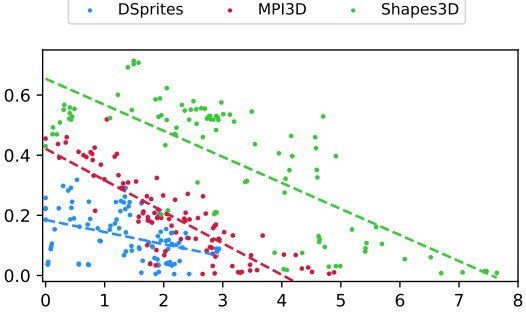

Figure 5: Alignment (y-axis) vs. $\hat{I}(Z_\alpha; N_\alpha)$ (x-axis).

### 5.3. Does our regularization term effectively increase the alignment level?

To analyze this, we randomly select, for each dataset, 10 of the 100 scenarios above and we train the encoders for different values of $\beta$. In all of them we set the temperature fixed to 0.01. We show in Figure 6 the value of different measures for different values of $\beta$. We can extract the following conclusions from this: (i) lower values of $\beta$ retain better the essence (Fig. 6a), since increasing $I(Z_\alpha; Y)$ prevails over decreasing $I(Z_\alpha; N_\alpha)$; (ii) lower values of $\beta$ also tend to retain more nuisances, since more entropic representations are encouraged in this case (Fig. 6b); (iii) higher values of

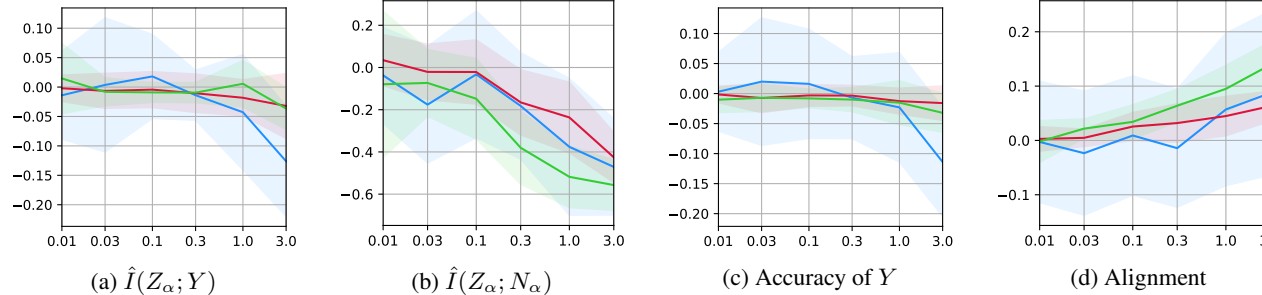

(a) $\hat{I}(Z_\alpha;Y)$  (b) $\hat{I}(Z_\alpha;N_\alpha)$  (c) Accuracy of $Y$  (d) Alignment

Figure 6: Relative change (with respect to the case in which $\beta = 0$) of different measures (y-axis) vs. $\beta$ (x-axis). The temperature is equal to 0.01 in all the cases. Same legend as Figure 5.

$\beta$ remove more nuisances, since they favor the decreasing of $I(Z_\alpha;N_\alpha)$ (Fig. 6b); (iv) higher values of $\beta$ also tend to discard more information of the essence, since they promote less entropic representations (Fig. 6a); (v) representations with lower $I(Z_\alpha;Y)$ result in a lower accuracy (Figs. 6a and 6c), as stated in Theorem 1; and (vi) representations with lower $I(Z_\alpha;N_\alpha)$ result in a higher alignment (Figs. 6b and 6d), as stated in Theorem 2.

**On the *Information Homeostasis* of the representations** In the previous experiment the temperature has been set fixed. However, as shown in Figure 4, lower temperatures tend to preserve more information of the nuisances. Thus, the next question arises: *Will the temperature be affected by the value of $\beta$?* To answer it, we repeat the previous experiment but setting the temperature as a trainable parameter. We show the results in Figure 7 and we observe that: (i) temperature tends to become lower when higher values of $\beta$ are used (Fig. 7a); and (ii) this translates into the fact that nuisances tend not to be eliminated to the same extent as in the case in which the temperature is fixed (Fig. 6b vs. 7b). We call this phenomenon *Information Homeostasis*, since it seems that, when an external stimulus (i.e., increasing $\beta$) affects the encoder, this activates available mechanisms (i.e., decreasing the temperature) in order to preserve to the extent possible the entropy of the representations (DelMonte & Kim, 2011). This effect becomes more pronounced for the highest values of $\beta$. In these cases, the level of nuisances is similar to the case in which $\beta = 0$, which reminds of an *homeostatic range* (Kotas & Medzhitov, 2015). This is an intriguing phenomenon that is out of the scope of this work.

## 6. Real-World Applications

Several real-world applications benefits from aligned representations. These include those that consist in generating one modality from another. We analyze what are the implications of introducing our regularization term in a real-world scenario. Concretely, we train a Q-Former with a frozen decoder-based LLM (Li et al., 2023) with different loss functions. In all the cases, two terms are present: (i) an image-text contrastive loss (ITC), i.e., the InfoNCE loss

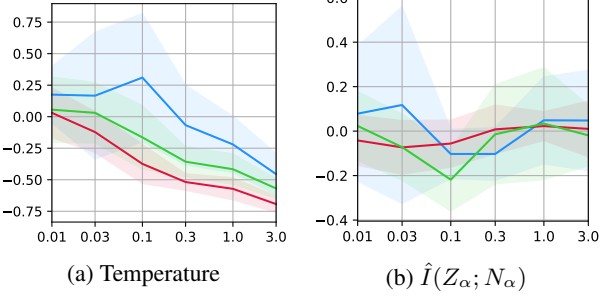

(a) Temperature  (b) $\hat{I}(Z_\alpha;N_\alpha)$

Figure 7: Relative change (with respect to the case in which $\beta = 0$) of different measures (y-axis) vs. $\beta$ (x-axis). Temperature is a trainable parameter. Same legend as Figure 5.

between image and text representations; and (ii) a language model loss (LM), which trains the Q-Former to generate text through the LLM using images as the condition. However, none of these losses explicitly encourages a high representational alignment. Thus, we experiment by adding: (i) an image-text matching loss (ITM), which is the binary cross-entropy loss of a task in which the model must predict if an image-text pair is positive or negative (Chen et al., 2020b); or (ii) our regularization term in equation (20). We note that, in contrast to ITM, our regularization term is modality-agnostic, computationally efficient and straightforward to implement. COCO (Lin et al., 2014) is used to train and test our model. More details are given in Appendix C.

### 6.1. Image Captioning

We argue that, for optimal performance in image captioning, the learned image representations should contain as little information as possible about nuisance factors. When nuisance information is retained, representations may encode fine-grained visual details that the text decoder is not equipped to handle, as it has not been trained to exploit such information. Table 3 summarizes the performance of different models. We observe the following trends: (i) loss functions that promote alignment between modalities tend to improve image captioning performance; (ii) there is a trade-off between image captioning and retrieval: caption-

Table 3: CIDEr (Vedantam et al., 2015), BLEU@4 (Papineni et al., 2002) and retrieval accuracy for Q-Formers trained with different loss functions.

| | CIDEr | BLEU@4 | I2T R@1 | T2I R@1 |
|---|---|---|---|---|
| ITC+LM | $91.7 \pm 0.2$ | $28.6 \pm 0.1$ | $\mathbf{64.2 \pm 0.2}$ | $\mathbf{52.3 \pm 0.4}$ |
| ITC+LM+ITM | $91.8 \pm 0.5$ | $28.8 \pm 0.2$ | $61.4 \pm 0.6$ | $49.7 \pm 0.8$ |
| ITC+LM+$0.01\mathcal{L}_M$ | $92.3 \pm 0.8$ | $29.1 \pm 0.4$ | $64.0 \pm 0.3$ | $\mathbf{52.3 \pm 0.5}$ |
| ITC+LM+$0.03\mathcal{L}_M$ | $92.6 \pm 0.3$ | $29.2 \pm 0.2$ | $63.9 \pm 0.4$ | $52.1 \pm 0.5$ |
| ITC+LM+$0.1\mathcal{L}_M$ | $\mathbf{93.0 \pm 0.3}$ | $\mathbf{29.4 \pm 0.3}$ | $63.0 \pm 0.5$ | $50.4 \pm 0.5$ |
| ITC+LM+$0.3\mathcal{L}_M$ | $90.5 \pm 0.4$ | $28.5 \pm 0.2$ | $59.6 \pm 0.4$ | $47.1 \pm 0.4$ |

ing benefits from minimal representations, whereas sufficient representations favors retrieval (as it is a downstream task); (iii) our loss slightly improves captioning performance for low values of $\beta$, with minimal degradation in retrieval performance; (iv) for moderate values of $\beta$, our loss substantially enhances captioning performance, though at a higher cost to retrieval accuracy; and (v) at high values of $\beta$, captioning and retrieval performances drop sharply, as representations become overly compressed and fail to retain sufficient task-relevant information—mirroring the trend found in Figure 6.

We also show in Figure 8 and in Appendix D some of the captions generated by the different models in Table 3. We observe that, in those cases in which representations are less aligned, captions tend to be more entropic because the representations are as well. This, in some cases, translates into captions that have information that does not correspond to the image. From a geometric point of view, we believe that in a misaligned space, for example, the representations of "lots of trees" and "a tree" are closer in the image than in the text space and, thus, the text decoder "confuses" them.

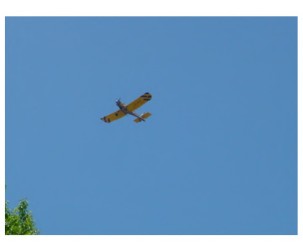 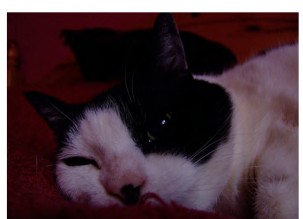

(1): a small airplane flying over a forest with lots of trees
(2): a small airplane flying through a blue sky above trees
(3): a small plane flying through a blue sky above a tree
(5): a small plane flying through a blue sky above a tree

(1): a cat is laying down on a bed with a white blanket
(2): a black and white cat laying on a red blanket
(3): a black and white cat laying on a red blanket
(5): a black wnd white cat laying on a bed

Figure 8: Captions generated by some of the trained models. Numbers correspondence is the same as in Table 3.

### 6.2. Multimodal Representation Space Arithmetic

Figure 9 shows image retrievals obtained from combining image and text representations from the Q-Former trained

with $\beta = 0.01$. Examples including other loss functions are found in Appendix E, showing that those not encouraging alignment, result in worse multimodal retrievals.

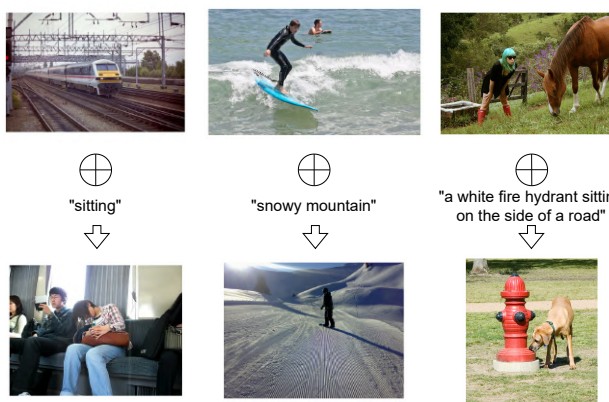

"sitting"

"snowy mountain"

"a white fire hydrant sitting on the side of a road"

Figure 9: Multimodal image retrieval for $\beta = 0.01$.

## 7. Related Work

**Information Bottleneck and Contrastive Representation Learning** Other works have previously connected IB to CRL, especially in the context of multi-view learning. In (Tian et al., 2020b), the authors argue that in CRL good views for a given task are those that optimize an IB w.r.t. that given task. Federici et al. (2020), analogously to us, propose a loss function to obtain representations that retain only the information shared by the two views, which is considered to be the relevant for downstream tasks. Tsai et al. (2020) build on the previous one and argue that including a reconstruction loss encourages to preserve the downstream task-relevant information. In (Wang et al., 2022) it is argued that, in the multi-view setting, imposing a strong IB can be detrimental for downstream tasks, since it could be removing an excess of information in the representation. However, to the best of our knowledge, the use of the IB in the multimodal setting remains understudied and the present is the first work that explores this and connects it to the misalignment typically present between representations from different modalities.

**Alignment of Multimodal Contrastive Learned Representations**  This is a well-studied field but still marked by a great amount of unanswered questions. The first work that extensively studied the alignment in CRL was (Wang & Isola, 2020) and demonstrated that, under infinite negative samples, the InfoNCE is globally minimized if the representations are perfectly aligned. They also define the representational alignment as in equation (17), which makes our formulation consistent with this work. However, it was observed in (Liang et al., 2022) that there exists in practice a great misalignment between representations from different modalities. This misalignment becomes problematic for tasks that need to combine both modalities. For example, Chen et al. (2020b); Li et al. (2021; 2022; 2023) use modifications to the InfoNCE loss to obtain a better performance in tasks in which image representations serve to obtain text, such as Image Captioning or Visual Question Answering. In the the opposite direction, Ramesh et al. (2022) use a generative model to transform text representations to image representations to train a text-conditioned image generator. In our view, this generative model serves to increase the entropy of the text representations to balance them with the image representations, which are typically more entropic. To the best of our knowledge, (Schrodi et al., 2024) is the first work in which this phenomenon is explained through the lens of an information imbalance. However, this imbalance is analyzed in the input space rather than in the representation space. Thus, aspects such as the encoder depth and hyperparameters or modifications of the loss function are not explored.

## 8. Conclusions

We give an explanation to the phenomenon of multimodal misalignment that usually emerges in encoders trained to minimize contrastive losses. These are designed to obtain representations that preserve the information about what is common to both modalities, but not to remove modality-specific information. We theoretically and empirically show that the presence of this modality-specific information in the representations is correlated with the misalignment phenomenon. We also examine the impact that different hyperparameters such as the temperature or encoder depth have on how much of this modality-information is removed. We derive a term that can be added to the contrastive loss which aims to eliminate this modality-specific information and, thus, allows to obtain a more aligned representation space. We find a phenomenon that we call *Information Homeostasis*, which consists in the fact that encoders seem to prefer representations with more nuisances and they modify, if possible, some of their internal parameters for this purpose. Finally, we show that our term in the loss function translates into a better performance in image captioning and seems to result in more consistent multimodal image retrievals.

## Acknowledgements

This work has received funding from the European Union's Horizon 2020 research and innovation programme under the Marie Skłodowska-Curie grant agreement No 101007666, MCIN/AEI/10.13039/501100011033 under Grant PID2021-126061OB-C44, and the Government of Aragón (Grant Group T36 23R). We are also grateful to the French National Research Agency for their support through the ANR-20-CE23-0012-01 (MIM) grant.

This work originated during the JSALT 2024 workshop. We gratefully acknowledge the researchers, mentors, and collaborators who took part in the workshop, as well as the staff at the Center for Language and Speech Processing at Johns Hopkins University, for cultivating a collaborative and intellectually rich environment that was instrumental in shaping this research. The workshop was partially supported by generous contributions from Amazon, Facebook, Google, and Microsoft.

## Impact Statement

This paper presents work whose goal is to advance the field of Machine Learning. There are many potential societal consequences of our work, none which we feel must be specifically highlighted here.

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

# A. Proofs of Sections 3 and 4

### A.1. Proof of Lemma 1

**Lemma 1.** *Let $Y$ and $Y'$ be essences of the same pair of modalities. Then, there exist a one-to-one transformation $\Psi$ such that $Y = \Psi(Y')$.*

*Proof.* By Definition 1, $Y$ and $Y'$ are minimal sufficient statistics of $X_\alpha$ for $X_\beta$. Then, by the definition of minimal sufficient statistic, there exist two functions $\Psi$ and $\Psi'$ such that $Y = \Psi(Y')$ and $Y' = \Psi'(Y)$. Then, $\Psi^{-1} = \Psi'$. $\quad\square$

### A.2. Proof of Theorem 1

**Theorem 1.** *Let $Y$ and $Z_\alpha$ be the essence and a representation of input $X_\alpha$ respectively, and let $\mathcal{T} = \{T : T = f(Y)\}$ be the set of deterministic functions of $Y$ (i.e., all the tasks derived from $Y$). Then, we have that:*

$$p(t|z_\alpha) = p(t|x_\alpha) \; \forall \, T \in \mathcal{T} \implies I(Z_\alpha; Y) = I(X_\alpha; Y)$$

*Proof.* First, we know from equation (4) that $I(Y; Z_\alpha) \leq I(Y; X_\alpha)$. Second, since $T = f(Y)$, we have the Markov Chain $Z_\alpha \leftrightarrow Y \leftrightarrow X_\alpha$ and, thus, by the DPI, $I(T; Z_\alpha) \leq I(Y; Z_\alpha)$. Third, since $p(t|x_\alpha) = p(t|z_\alpha)$, we know that $I(T; X_\alpha) = I(T; Z_\alpha)$. Thus, we have that $I(T; X_\alpha) = I(T; Z_\alpha) \leq I(Y; Z_\alpha) \leq I(Y; X_\alpha)$. Finally, since $T$ can be any function of $Y$, it can be the identity function, in which case $I(T; X_\alpha) = I(Y; X_\alpha)$. $\quad\square$

### A.3. Proof of Theorem 2

**Theorem 2** (Informal). *Let $Z_\alpha$ and $Z_\beta$ be two representations of a pair of inputs with nuisances $N_\alpha$ and $N_\beta$ respectively, such that $Z_\alpha$ and $Z_\beta$ are aligned in the sense of equation (2). Then, $I(Z_\alpha; N_\alpha) = I(Z_\beta; N_\beta) = 0$.*

*Proof.* First, if $I(Z_\alpha; N_\alpha) \neq 0$, then there exists a surjective function $f$ such that $Z_\beta = f(Z_\alpha)$, i.e., more than one representation of modality $\alpha$ can correspond with one representation of modality $\beta$. Let $\left\{ z_\alpha^{(l)} \sim p_{\theta_\alpha}\left(z|x_\alpha^{(l)}\right) : x_\alpha^{(l)} \sim p(x_\alpha) \right\}$ and $\left\{ z_\beta^{(l)} \sim p_{\theta_\beta}\left(z|x_\beta^{(l)}\right) : x_\beta^{(l)} \sim p(x_\beta) \right\}$ be two infinite sets. Then, there exists a pair $(l, l')$ for which $z_\alpha^{(l)} \neq z_\alpha^{(l')}$ and $z_\beta^{(l)} = z_\beta^{(l')}$ and for which $K(z_\alpha^{(l)}, z_\alpha^{(l)}) \neq K(z_\alpha^{(l)}, z_\alpha^{(l')})$, but $K(z_\beta^{(l)}, z_\beta^{(l)}) = K(z_\beta^{(l)}, z_\beta^{(l')})$. $\quad\square$

We must note that, in practice, we usually work with finite sets, so we could obtain the maximum value of alignment while having the presence of nuisances in the representation.

### A.4. Proof of $I(Z_\alpha; Y) = I(Z_\alpha; X_\beta)$

*Proof.*

$$p(z_\alpha|y, x_\beta) = \int p(z_\alpha|y, x_\beta, x_\alpha) p(x_\alpha|y, x_\beta) \, dx_\alpha \tag{21}$$

$$= \int p(z_\alpha|x_\alpha) p(x_\alpha|y) \, dx_\alpha = p(z_\alpha|y) \tag{22}$$

In line (21), we apply Definition 3 and equation (4). Then, we have the following Markov Chain $X_\beta \leftrightarrow Y \leftrightarrow Z_\alpha$ and, by the DPI, $I(Z_\alpha; Y) \geq I(Z_\alpha; X_\beta)$.

$$p(z_\alpha|y, x_\beta) = \int p(z_\alpha|y, x_\beta, x_\alpha) p(x_\alpha|y, x_\beta) \, dx_\alpha \tag{23}$$

$$= \int p(z_\alpha|x_\alpha) p(x_\alpha|x_\beta) \, dx_\alpha = p(z_\alpha|x_\beta) \tag{24}$$

In line (23), we apply Definition 3 and equation (5). Then, we have the following Markov Chain $Y \leftrightarrow X_\beta \leftrightarrow Z_\alpha$ and, by the DPI, $I(Z_\alpha; Y) \leq I(Z_\alpha; X_\beta)$. $\quad\square$

### A.5. Proof of equation (17)

*Proof.*

$$I(Z_\alpha; X_\alpha) = \iint p_{\theta_\alpha}(z, x_\alpha) \log \frac{p_{\theta_\alpha}(z|x_\alpha)}{p_{\theta_\alpha}(z)} \, dz \, dx_\alpha \tag{25}$$

$$= \iiint p_{\theta_\alpha}(z|x_\alpha) p(x_\alpha, x_\beta) \log \frac{p_{\theta_\alpha}(z|x_\alpha)}{p_{\theta_\alpha}(z)} \, dz \, dx_\alpha \, dx_\beta \tag{26}$$

$$= \iiint p_{\theta_\alpha}(z|x_\alpha) p(x_\alpha, x_\beta) \log \frac{p_{\theta_\alpha}(z|x_\alpha)}{p_{\theta_\beta}(z|x_\beta)} \frac{p_{\theta_\beta}(z|x_\beta)}{p_{\theta_\alpha}(z)} \, dz \, dx_\alpha \, dx_\alpha \tag{27}$$

$$= \mathop{\mathbb{E}}_{p(x_\alpha, x_\beta)} \left[ D_{KL} \left( p_{\theta_\alpha}(z|x_\alpha) || p_{\theta_\beta}(z|x_\beta) \right) \right] - \mathop{\mathbb{E}}_{p(x_\alpha, x_\beta|z)} \left[ D_{KL} \left( p_{\theta_\alpha}(z) || p_{\theta_\beta}(z|x_\beta) \right) \right] \tag{28}$$

$$\leq \mathop{\mathbb{E}}_{p(x_\alpha, x_\beta)} \left[ D_{KL} \left( p_{\theta_\alpha}(z|x_\alpha) || p_{\theta_\beta}(z|x_\beta) \right) \right] \tag{29}$$

$\square$

### A.6. Proof of equation (18)

*Proof.* Let $p_{\theta_\alpha}(z \mid x_\alpha) = \mathcal{N}(z; \mu_\alpha, \sigma^2 I)$ and $p_{\theta_\beta}(z \mid x_\beta) = \mathcal{N}(z; \mu_\beta, \sigma^2 I)$, with $\mu_\alpha = \mu_{\theta_\alpha}(x_\alpha)$ and $\mu_\beta = \mu_{\theta_\beta}(x_\beta)$. The KL divergence between these two Gaussians is given by:

$$D_{\mathrm{KL}}(p \,\|\, q) = \frac{1}{2} \left[ \mathrm{tr}(\Sigma_q^{-1} \Sigma_p) + (\mu_q - \mu_p)^\top \Sigma_q^{-1}(\mu_q - \mu_p) - d + \log \frac{\det \Sigma_q}{\det \Sigma_p} \right] \tag{30}$$

where $p = \mathcal{N}(\mu_p, \Sigma_p)$ and $q = \mathcal{N}(\mu_q, \Sigma_q)$. For $\Sigma_p = \Sigma_q = \sigma^2 I$, Eq. (30) simplifies to:

$$D_{\mathrm{KL}} \left( p_{\theta_\alpha}(z \mid x_\alpha) \,\|\, p_{\theta_\beta}(z \mid x_\beta) \right) = \frac{1}{2\sigma^2} \left\| \mu_{\theta_\alpha}(x_\alpha) - \mu_{\theta_\beta}(x_\beta) \right\|_2^2. \tag{31}$$

Taking expectation over the joint distribution $p(x_\alpha, x_\beta)$, we obtain:

$$\mathbb{E}_{p(x_\alpha, x_\beta)} \left[ D_{\mathrm{KL}} \left( p_{\theta_\alpha}(z \mid x_\alpha) \,\|\, p_{\theta_\beta}(z \mid x_\beta) \right) \right] = \frac{1}{2\sigma^2} \mathbb{E}_{p(x_\alpha, x_\beta)} \left[ \left\| \mu_{\theta_\alpha}(x_\alpha) - \mu_{\theta_\beta}(x_\beta) \right\|_2^2 \right] \tag{32}$$

Hence, the expected KL divergence is proportional to the expected squared $\ell_2$ distance between the mean embeddings:

$$\mathbb{E}_{p(x_\alpha, x_\beta)} \left[ D_{\mathrm{KL}} \left( p_{\theta_\alpha}(z \mid x_\alpha) \,\|\, p_{\theta_\beta}(z \mid x_\beta) \right) \right] \propto \mathbb{E}_{p(x_\alpha, x_\beta)} \left[ \left\| \mu_{\theta_\alpha}(x_\alpha) - \mu_{\theta_\beta}(x_\beta) \right\|_2^2 \right]. \tag{33}$$

The constant of proportionality is $\frac{1}{2\sigma^2}$ and independent of the model parameters $\theta_\alpha, \theta_\beta$. $\square$

## B. Connection between our loss function and temperature

In the case where the embeddings are unit-norm, our proposed loss takes the form:

$$\mathcal{L}_i = \log \frac{\exp(s_{ii}/\tau)}{\sum_k \exp(s_{ik}/\tau)} + 2\beta(1 - s_{ii}), \tag{34}$$

where $s_{ik}$ denotes the cosine similarity between the embeddings $z^{(i)}$ and $z^{(k)}$. The gradient of $\mathcal{L}_i$ with respect to each similarity term is given by:

$$\frac{\partial \mathcal{L}_i}{\partial s_{ii}} = -\frac{1}{\tau}\left(1 - \frac{\exp(s_{ii}/\tau)}{\sum_k \exp(s_{ik}/\tau)}\right) - 2\beta, \tag{35}$$

$$\frac{\partial \mathcal{L}_i}{\partial s_{ij}} = \frac{1}{\tau} \cdot \frac{\exp(s_{ij}/\tau)}{\sum_k \exp(s_{ik}/\tau)} \quad \text{for } j \neq i. \tag{36}$$

We further analyze a modified variant of the InfoNCE loss where the temperature differs between the numerator and denominator:

$$\mathcal{L}'_i = \log \frac{\exp(s_{ii}/\tau')}{\sum_k \exp(s_{ik}/\tau)}. \tag{37}$$

The gradients of this variant are:

$$\frac{\partial \mathcal{L}'_i}{\partial s_{ii}} = -\frac{1}{\tau'}\left(1 - \frac{\exp(s_{ii}/\tau')}{\sum_k \exp(s_{ik}/\tau)}\right), \tag{38}$$

$$\frac{\partial \mathcal{L}'_i}{\partial s_{ij}} = \frac{1}{\tau} \cdot \frac{\exp(s_{ij}/\tau)}{\sum_k \exp(s_{ik}/\tau)} \quad \text{for } j \neq i, \tag{39}$$

which matches Eq. (36), confirming that the two losses only differ in their treatment of $s_{ii}$.

By comparing Eq. (35) and Eq. (38), we find that our regularized loss is equivalent to optimizing a variant of InfoNCE with a temperature mismatch between numerator and denominator. Solving for $\beta$, we obtain:

$$\beta = \frac{1}{2}\left[\frac{\tau - \tau'}{\tau\tau'} + \frac{\exp(s_{ii}/\tau) - \exp(s_{ii}/\tau')}{\sum_k \exp(s_{ik}/\tau)}\right]. \tag{40}$$

This expression reveals that:

- The effective temperature gap depends on the similarity between the anchor and all other samples in the batch.

- When $s_{ii} \ll \sum_k \exp(s_{ik}/\tau)$, i.e., predictions are far from the target distribution, the second term in Eq. (40) is negligible, and

$$\beta \approx \frac{1}{2} \cdot \Delta\tau$$

  with $\Delta\tau = \frac{\tau - \tau'}{\tau\tau'}$. Thus, larger values of $\beta$ correspond to larger temperature mismatches.

- Conversely, when $s_{ii} \approx \sum_k \exp(s_{ik}/\tau)$, i.e., the model is confident in its match, we have:

$$\beta \approx \frac{1}{2}\left[\Delta\tau + 1 - \exp(\Delta\tau)\right],$$

  indicating that the temperature gap required to match a fixed $\beta$ is smaller in this regime.

Hence, our regularizer can be interpreted as an adaptive temperature adjustment that decreases the denominator temperature when the model is uncertain, and aligns it closer to the numerator temperature when predictions are confident.

## C. Experimental Details

Below, we provide a summary of the experimental setup. Full implementation details and code are available at: `https://github.com/antonioalmudevar/multimodal_ib`.

Table 4: Hyperparameters of Section 5

|  | DSprites | MPI3D | Shapes3D |
|---|---|---|---|
| factors encoder MLP | $\{16, 16, 8, 8, 16, 16\}$ | $\{128, 128, 64, 64, 128, 128\}$ | $\{64, 64, 32, 32, 64, 64\}$ |
| number of epochs | 50 | 50 | 50 |
| batch size | 128 | 128 | 128 |
| optimizer | Adam | Adam | Adam |
| learning rate | 0.001 | 0.001 | 0.001 |
| scheduler | Step | Step | Step |
| step size (epochs) | 20 | 20 | 20 |
| scheduler $\gamma$ | 0.3 | 0.3 | 0.3 |

Table 5: Categories of factors in section 5.1. In MPI3D, the factor background_color actually refers to the color of a ring in the images, so we consider it as an object (see `https://github.com/rr-learning/disentanglement_dataset`). There are some factors missing because the do not fall into any category.

|  | DSprites | MPI3D | Shapes3D |
|---|---|---|---|
| Location | posX, posY | horizontal_axis, vertical_axis | orientation |
| Shape | shape | object_shape | shape |
| Size | size | object_size | scale |
| Object Color | - | object_color, background_color | object_hue |

Table 6: Hyperparameters of Section 6

| | |
|---|---|
| vision encoder | VIT-g/14 (Fang et al., 2023) |
| image size | 224 |
| # of query tokens | 32 |
| cross attention frequency | 2 |
| representation dimension | 256 |
| text encoder | BERT$_{base}$(Devlin, 2018) |
| batch size | 128 |
| optimizer | Adam |
| learning rate | 0.0001 |
| optimizer $\beta$ | $(0.9, 0.999)$ |
| scheduler | cosine annealing |
| warm-up steps | 1000 |
| training steps | 50000 |

## D. More Results of Section 6.1

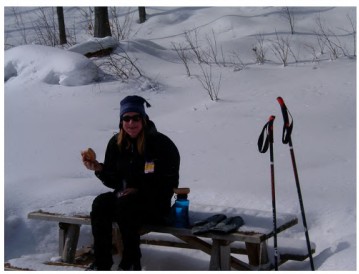

(1): a woman sitting on a bench with a bag of food
(2): a woman sitting on a bench with a plate of food
(3): a woman sitting on a bench in the snow
(5): a woman sitting on a bench in the snow

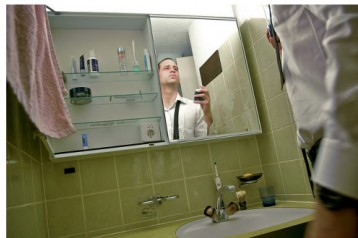

(1): a man is taking a picture of himself in the mirror
(2): a man is taking a picture of himself in the mirror
(3): a man is seen in the reflection of a mirror
(5): a man standing in front of a bathroom mirror

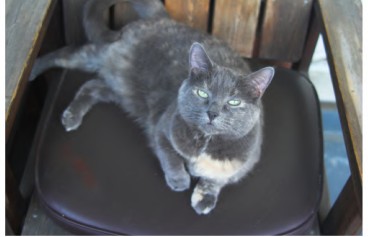

(1): a cat laying on top of a wooden chair in a room
(2): a cat sitting on a chair looking at the camera
(3): a cat sitting on a chair looking at the camera
(5): a cat laying on top of a wooden chair

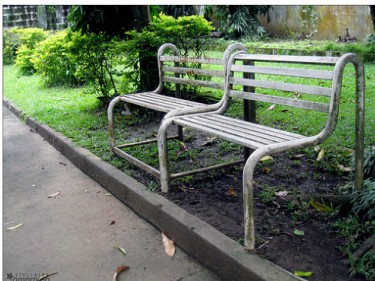

(1): a couple of benches that are in the grass
(2): a white park bench sitting next to a tree
(3): a bench sitting in the middle of a park
(5): a metal bench sitting in the middle of a park

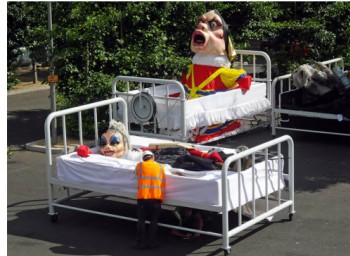

(1): a toy set of a man in a hospital bed with a robot in the middle
(2): two dummy heads are on a bed that is shaped like a boat
(3): a fake bed with a dummy head and legs on it
(5): a display of a demonic joker character in a bed

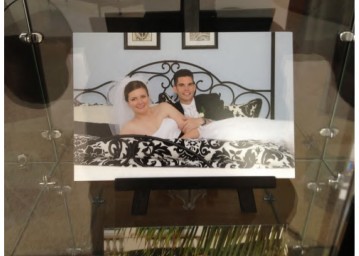

(1): a man and woman in a white dress are sitting on a bed
(2): a man and woman are sitting in a bed
(3): a picture of a couple in a bedroom with a glass frame
(5): a photo of a couple in a frame on a bed

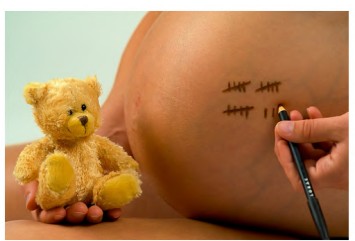
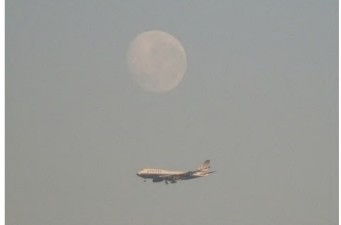
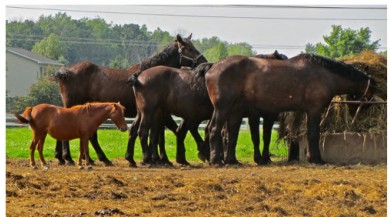

(1): a person holding a teddy bear with a pair of scissors
(2): a person holding a teddy bear with a woman's pregnant belly
(3): a person with a teddy bear on their lap
(5): a pregnant woman holding a teddy bear with a baby on it

(1): a jetliner flying through a cloudy blue sky
(2): a plane flying in the sky with a half moon in the background
(3): a plane flying in the sky with a half moon in the background
(5): a plane flying in the sky with a half moon in the background

(1): a group of horses standing around a baby horse
(2): a group of horses standing around a pile of hay
(3): a group of horses eating hay in a field
(5): a group of horses standing next to each other

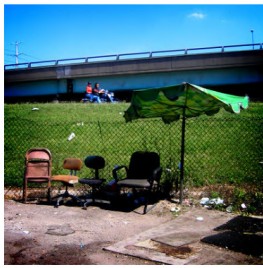
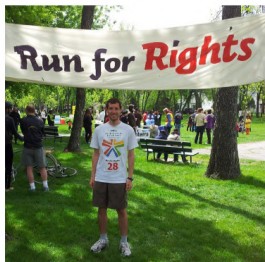
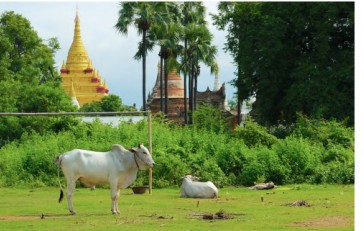

(1): a couple of chairs and an umbrella on a field
(2): a couple of chairs sitting next to a table
(3): a couple of chairs and a table on a field
(5): a couple of chairs sitting next to a fence

(1): a man standing under a banner with a frisbee
(2): a man standing under a banner that says bicycle fair
(3): a man standing under a banner that says bicycle relief
(5): a man standing under a sign that reads for a cause

(1): a cow standing in a field with a house in the background
(2): a cow is standing in a field with a house in the background
(3): a couple of cows standing on top of a lush green field
(5): a cow is standing in a field with a few trees

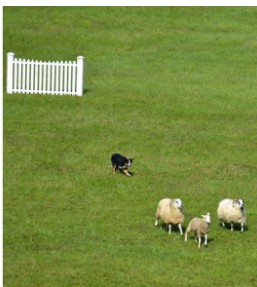
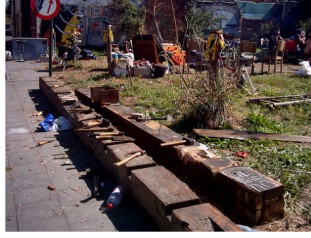
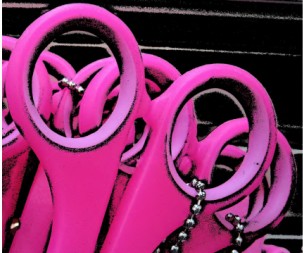

(1): a dog herding sheep in a field with a dog
(2): a dog herding sheep in a field with a dog running behind them
(3): a dog is herding three sheep in a field
(5): a dog is herding some sheep in a field

(1): a wooden bench sitting in the middle of a yard
(2): a pile of wood sitting on the side of a road
(3): a pile of wood sitting on the side of a road
(5): a bunch of old used wooden poles in a yard

(1): a close up of a pair of scissors in a pile
(2): a group of scissors that are hanging up together
(3): a bunch of pink scissors are chained together in a room
(5): a bunch of pink scissors are all grouped up

Figure 10: Captions generated by some of the trained models. Numbers correspondence is the same as in Table 3.

# E. More Results of Section 6.2

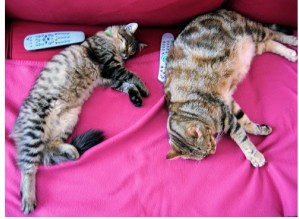 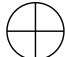 "a woman with her face close to a mans face"

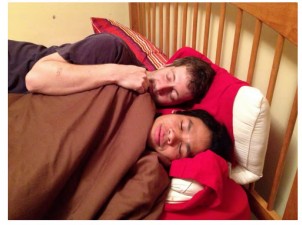 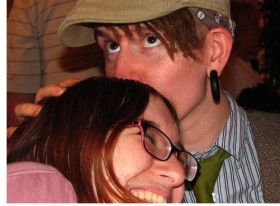 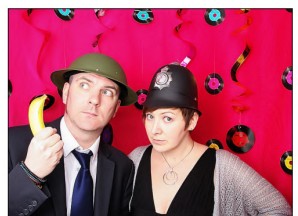 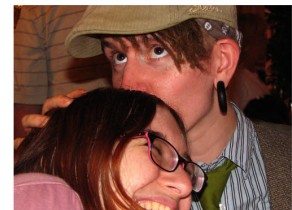

ITC+LM      ITC+LM+ITM      ITC+LM+$0.01\mathcal{L}_M$      ITC+LM+$0.1\mathcal{L}_M$

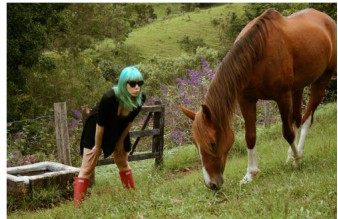 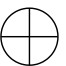 "a white fire hydrant sitting on the side of a road"

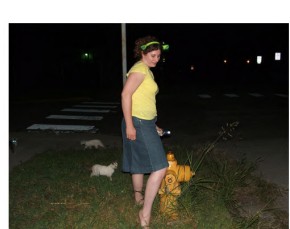 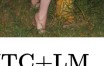 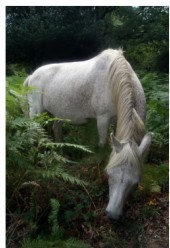 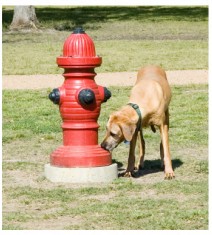 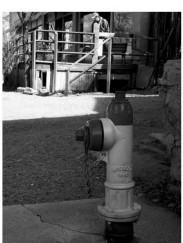

ITC+LM      ITC+LM+ITM      ITC+LM+$0.01\mathcal{L}_M$      ITC+LM+$0.1\mathcal{L}_M$

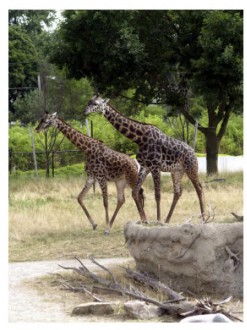 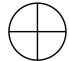 "snowy mountain"

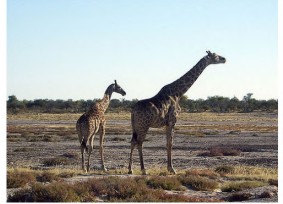 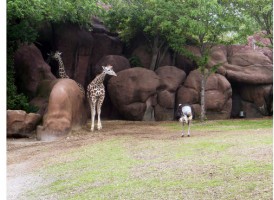 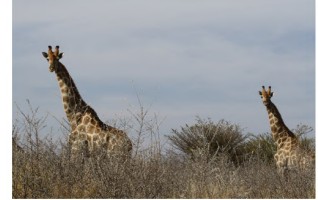 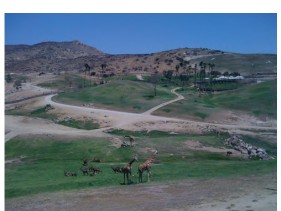

ITC+LM        ITC+LM+ITM        ITC+LM+$0.01\mathcal{L}_M$        ITC+LM+$0.1\mathcal{L}_M$

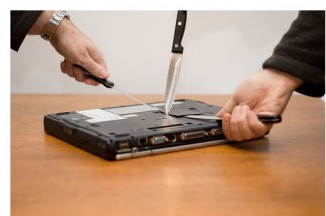 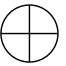 "modern building"

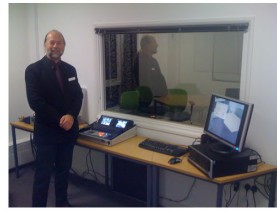 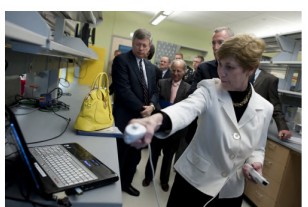 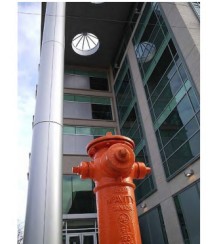 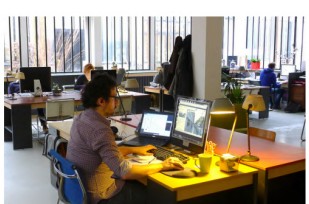

ITC+LM        ITC+LM+ITM        ITC+LM+$0.01\mathcal{L}_M$        ITC+LM+$0.1\mathcal{L}_M$

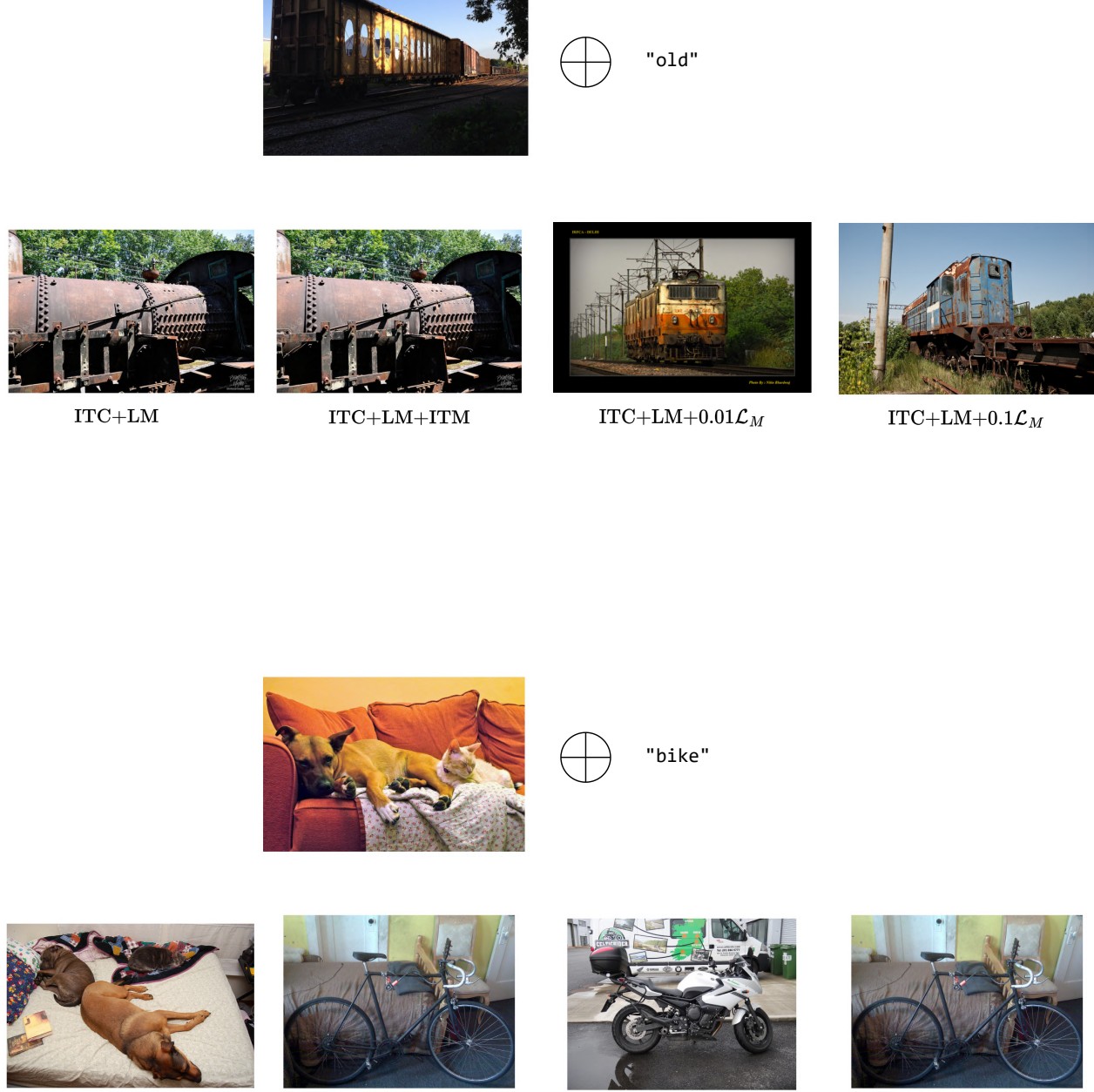

Figure 11: Multimodal image retrievals from models train with different loss functions. We believe that text representations from more simple captions (less entropic) are better aligned with image representations from encoders trained with a higher values of $\beta$, since they are less entropic too.

