# OpenReview forum: "Aligning Multimodal Representations through an Information Bottleneck"
_ICML.cc/2025/Conference — ICML 2025 poster_

### Official Review · Reviewer_N9GW · 2025-03-11

**Overall Recommendation:** 4

**Summary:**

In this paper, the authors study the alignment of representation in multimodal learning through information theory.
For a positive pair $X_\alpha, X_\beta$ from modalities $\alpha, \beta$, they formulate the essence $Y$ and nuisance
$N_\alpha, N_\beta$ of the inputs as the common and modality-specific parts (in the mutual information sense) of the
inputs, respectively. Then, they define a representation $Z_\alpha$ of $X_\alpha$ to be sufficient if it preserves all
information in $Y$ and say $Z_\alpha$ is minimal if it contains no information about $N_\alpha$. After that, they
relate these notions to the maximization of $I(Z_\alpha; Y)$ and minimization of $I(Z_\alpha; N_\alpha)$, and show that
minimizing a regularized version of InfoNCE can be used as a surrogate of this optimization task. Finally, they provide
experimental evidence supporting the validity of their formulation and the utility of their regularizer.

## update after rebuttal
I thank the authors for correcting my misunderstandings and the new discussion on the regularizer. I will keep my score.

**Claims And Evidence:**

* This paper proposes an information theoretical interoperation (essence, nuisance, and sufficient/minimal representations)
  of the misalignment phenomenon in multimodal learning. They provide both theoretical results and toy experiments to
  support their interpretation.
* Based on their interpretation, they propose a regularizer to reduce the amount of nuisance contained in the learned
  representation and verify on real-world datasets and it improves the performance of the model.

**Essential References Not Discussed:**

No

**Experimental Designs Or Analyses:**

Yes. The experiment designs and analyses are sound. For the real-world applications, it would be good to have experiments
on more recent datasets.

**Methods And Evaluation Criteria:**

* They verify thier interpretation on various toy datasets (DSprites, MPI3D, Shapes3D), which allow them to control the
  ratio between essence and nuisance in the data.
* They approximate the amount of preserved nuisance (which is not computatable in general) by training a linear classifier
  to predict the nuisance from the learned representation. This makes sense and is also used in previous works.
* For those more realistic experiments, they use CIDEr and BLEU@4 to test the performance of the regularized/unregularized
  models and provide some examples for the image retrieval task. I'm not familiar with the more empirical side of this
  field, but these datasets seem to be rather old (from 2015 and 2002).

**Other Comments Or Suggestions:**

Consider using the mathrm or texttt when writing, say, HSIC and infoNCE in an equation.

**Other Strengths And Weaknesses:**

Overall, this is a neat paper, well-written and easy-to-follow.

**Questions For Authors:**

If we normalize the output representation $\mu$ to have unit norm, then the $l^2$ distance between $\mu_\alpha$ and
$\mu_\beta$ is equivalent to the (negative) cosine similarity of them. Meanwhile, (at least when the temperature is high,)
we can linearize/Taylor expand the softmax in InfoNCE to get the cosine similarity out. Is it possible to combine these
to explain the model's behavior under the proposed regularization?

I'm asking this purely out of curiosity, and the response is unlikely to affect the score.

**Relation To Broader Scientific Literature:**

The key contribution of this paper is to extend the idea of (Tian et al., 2020b) to the multimodal setting, where the
definition of common and modality-specific parts of inputs are less clear, and based on this extension, provide an
explanation on the misalignment phenomenon in multimodal learning.

**Theoretical Claims:**

Yes. The theoretical claims are valid. However, the role of Theorem 1 is rather unclear. It says that if one can solve
all downstream tasks using a representation $Z_\alpha$, then it is sufficient in the sense of Definition 4. To justify
the definition, the reverse direction looks more natural.

---

> ### Author Rebuttal · Authors · 2025-03-31
>
> We would like to begin by thanking you for the time dedicated to give us feedback to improve our work. We address your main concerns next:
>
> > these datasets seem to be rather old (from 2015 and 2002).
>
> We believe that you may be refering to the metrics instead of to the datasets. These are still widely used despite of the fact that they are old. See for example [1] and [2], two recent papers with a great impact in the field.
>
> [1] Li, J., Li, D., Savarese, S., & Hoi, S. (2023, July). Blip-2: Bootstrapping language-image pre-training with frozen image encoders and large language models.
>
> [2] Team, G., L. (2023). Gemini: a family of highly capable multimodal models.
>
> > the role of Theorem 1 is rather unclear.
>
> Theorem 1 states that sufficiency is a necessary condition to solve all the downstream tasks (that can be derived from the essence). In other words, if a representation is not sufficient, then it cannot solve all the downstream tasks. The opposite is also true, but we do not find it so valuable. Having a representation that can potentially solve all the downstream tasks is not valuable because it does give no information on how “easy-to-use” the information to solve this task is. In the extreme case, the input can be seen as a representation of itself (since it satisfies Definition 3) and it can potentially (through an oracle model) solve any downstream task. Thus, obtaining a representation that can potentially solve all the downstream tasks is not meritorious nor theoretically valuable. However, the importance of this theorem lies in the fact that any representation that is not sufficient cannot be used to perfectly solve any downstream task, no matter how “easy-to-use” the information it contains is, which enhances the value of sufficient representations. This idea connects to that of usable information [3], which allows to formulate this differentiation between having information present in a representation and how “easy-to-use” this information is. This clarification can be made in the paper.
>
> [3] Xu, Y., Zhao, S., Song, J., Stewart, R., & Ermon, S. (2020). A theory of usable information under computational constraints. arXiv preprint arXiv:2002.10689.
>
> > Consider using the mathrm or texttt...
>
> mathrm will be used.
>
> > If we normalize the output representation $\mu$ ...
>
> In the case in which the embeddings are unit-norm, our loss becomes $\mathcal{L}_i = \log\frac{\exp(s\_{ii}/\tau)}{\sum_k \exp(s\_{ik}/\tau)} + 2\beta(1-s\_{ii})$, where $s\_{ik}$ is the cosine similarity between $z^{(i)}$ and $z^{(k)}$. We have the following:
> - $\frac{\partial{\mathcal{L}_i}}{\partial s\_{ii}} = -\frac{1}{\tau} \left(1 - \frac{\exp(s\_{ii}/\tau)}{\sum_k \exp(s\_{ik}/\tau)} \right) - 2\beta$
> - $\frac{\partial{\mathcal{L}_i}}{\partial s\_{ij}} = \frac{1}{\tau} \frac{\exp(s\_{ij}/\tau)}{\sum_k \exp(s\_{ik}/\tau)}$
>
> We also analyze the gradients of a modification of the $\mathrm{InfoNCE}$ in which a different temperature is used for the numerator and the denominator, i.e., $\mathcal{L}'_i=\log\frac{\exp(s\_{ii}/\tau')}{\sum_k \exp(s\_{ik}/\tau)}$. Then, it can be easily checked that:
> - $\frac{\partial{\mathcal{L}'_i}}{\partial s\_{ii}} = -\frac{1}{\tau'} \left(1 - \frac{\exp(s\_{ii}/\tau')}{\sum_k \exp(s\_{ik}/\tau)} \right)$
> - $\frac{\partial{\mathcal{L}'_i}}{\partial s\_{ij}} = \frac{\partial{\mathcal{L}_i}}{\partial s\_{ij}}$
>
> Thus, we have that optimizing our loss is equivalent to optimizing a modification of $\mathrm{InfoNCE}$ with different temperature in numerator and denominator. Rearranging, $\beta=\frac{1}{2} \left[ \frac{\tau-\tau'}{\tau\tau'} + \frac{\exp(s\_{ii}/\tau) - \exp(s\_{ii}/\tau')}{\sum_k \exp(s\_{ik}/\tau)} \right]$. Thus:
>
> - The difference in the temperature between numerator and denominator depends on the similarity with respect to all the elements in the batch.
> - If $s\_{ii} \ll \sum_k \exp(s\_{ik}/\tau)$, i.e., the predictions are far from the target distribution, then $\beta \approx \frac{1}{2}  \Delta\tau$, where $\Delta\tau = \frac{\tau-\tau'}{\tau\tau'}$. Thus, the larger the $\beta$, the larger is the difference between the numerator and denominator temperatures.
> - If $s\_{ii} \gg \sum_k \exp(s\_{ik}/\tau)$, i.e., the predictions are close to the target distribution, then $\beta \approx \frac{1}{2} \left[ \Delta\tau + 1 - \exp\Delta\tau \right]$. Thus, the temperature difference between numerator and denominator for a given value of $\beta$ is lower than in the previous case.
>
> Thus, our term can be seen as a regularizer the adapts the value of the temperature in the denominator based on the how close to the true distribution the prediction is.
>
> These comments will be added in an appendix. Thank you for your interest. Please let us know if you have any other insight with respect to this last analysis.
>
> We hope that your questions have been addressed. If this is the case and you consider that our paper deserves an increase in the score, we would thank you if you made it effective.

---

### Official Review · Reviewer_Xy8R · 2025-03-14

**Overall Recommendation:** 4

**Summary:**

The manuscript shows that contrastive learning methods for multimodal representations do not remove modality-specific information, which leads to misaligned representations. It uses an Information Bottleneck approach to add a regularization term to the loss function to filter out this extra information while preserving the alignment. The approach demonstrates improved performance in tasks such as image captioning and multimodal retrieval.

## update after rebuttal
The score was increased from 3 (Weak Accept) to 4 (Accept). Most of the concerns have been addressed. The authors reran the experiments multiple times and included additional experiments using different architectures to evaluate how architectural choices affect the URR. They also clarified the questions regarding nuisances, which resolved my earlier confusion.

**Claims And Evidence:**

- While the paper supports its claims about contrastive losses failing to remove nuisance information using the Uncertainty Reduction Ratio (URR), I have a few concerns. First, could the choice of image encoder introduce bias in the URR measurements? Second, how are invariance and equivariance with respect to different factors accounted for in this analysis? Finally, I recall an ICLR 2021 paper suggesting that neural networks tend to focus on the easiest factor to minimize the training objective rather than learning all factors. How does this observation affect the interpretation of the paper’s results on nuisance information removal?
-  Minimal sufficient representations argue that achieving representational alignment requires that the learned representations be both sufficient (containing all shared “essence”) and minimal (excluding nuisances). Hence, the manuscript introduces a regularization term for alignment. This seems to be supported by the results in Table 2.
- For the Information Homeostasis phenomenon, the evidence seems preliminary. Considering that InfoNCE employs a softmax-like objective, could this phenomenon be linked to issues inherent to softmax formulations, as discussed by Veličković et al. (2024)? Moreover, Zhai et al. (2023) suggest that using sigmoids instead for sigmoids.

Veličković, Petar, et al. "softmax is not enough (for sharp out-of-distribution)." arXiv preprint arXiv:2410.01104 (2024).
Zhai, Xiaohua, et al. "Sigmoid loss for language image pre-training." Proceedings of the IEEE/CVF international conference on computer vision. 2023.

**Essential References Not Discussed:**

I do not have concerns on this.

**Experimental Designs Or Analyses:**

Table 2 needs to show std by running repeated experiments with different seeds.

**Methods And Evaluation Criteria:**

The methods and evaluation criteria seem reasonable.

**Other Comments Or Suggestions:**

- I think preliminaries can be cut down to related work without the equations.
- Also, it would be great if you annotated the essence and nuisances in the examples.

**Other Strengths And Weaknesses:**

Strengths:
- Multiple datasets
- Real-world experiments
- Ablations for the hyperparameters

**Questions For Authors:**

See above. I will be open to increasing the score based on more clarification.

**Relation To Broader Scientific Literature:**

Broadly speaking, this work seems to be InfoNCE with regularization on the latent space to ensure that multimodal representation aligns, where latent space is VAE with Gaussian priors but without reconstruction.

**Theoretical Claims:**

I think Equation 18 should be derived in the appendix.

---

> ### Author Rebuttal · Authors · 2025-03-31
>
> We would like to begin by thanking you for the time dedicated to give us feedback to improve our work. We address your main concerns next:
>
> > could the choice of image encoder introduce bias in the URR measurements?
>
> To analyze this point, we have performed experiments that are identical to those in Table 1, but using a small ViT as image encoder. We show the results next:
>
> ||DSprites|MPI3D|Shapes3D|
> |-|-|-|-|
> |Location|$2.8\pm 0.6$|$2.8\pm 0.3$|$1.1\pm 0.1$|
> |Shape|$64.9\pm 1.4$|$7.0\pm 0.4$|$5.7\pm 0.2$|
> |Size|$30.7\pm 3.5$|$20.8\pm 3.5$|$6.9\pm 1.5$|
> |Objects Color|-|$63.5\pm 9.8$|$53.5\pm 1.6$|
>
> By comparing the latter and Table 1 we can observe that:
>
> 1. Local attributes, such as *Location* and *Size*, are better preserved in convolutional encoders, which is consistent with the inductive biases towards local structures of convolutional layers.
> 2. Global attributes, such as *Shape* and *Objects Color* are almost equally conserved in both image encoders.
>
> Subsection 5.1 would be extended with these two experiments and conclusions. Thanks for noticing.
>
> > how are invariance and equivariance with respect to different factors accounted for in this analysis?
>
> Sorry, we do not understand this question, could you please further develop it?
>
> > I recall an ICLR 2021 paper...
>
> We think that the paper that you could be referring to is [1]. If this is the case, your question is very interesting and points of [1] and ours perfectly compatible.
>
> On the one hand, what the mentioned paper states is that, if the task to be solved is correlated with other simpler features, then a model trained with SGD will tend to learn these simpler features rather than the task. If we relate this idea with the concepts used in our work, the point of [1] would be something like: “given a task Y, models trained with SGD tend to learn minimal sufficient statistics of Y instead of Y”. For example, if our task were to classify between images of bananas and strawberries, our model would simply learn to differentiate between yellow and red.
>
> On the other hand, our paper argues that, even given the previous statement, models tend not to remove all the information that is unnecessary to solve the task (i.e., nuisances). In the previous example, a model could be learning, for instance, the background color of the images. Thus, our model could learn the fruit color and the background color, which makes the point of [1] and ours perfectly compatible.
>
> If this is the paper you refered to, thanks for bringing it up, its results are very interesting. If this is not the case, we are open to discuss other works.
>
> [1] Ahmed, F., Bengio, Y., Van Seijen, H., & Courville, A.. Systematic generalisation with group invariant predictions.
>
> > For the Information Homeostasis..., could this phenomenon be linked to issues inherent to softmax formulations?
>
> First, for higher values of $\beta$, representations tend not to retain the information that is unique to their input. Then, all the representations are more similar to each other and closer in the space. Thus, maybe predictions are less sharp, so the temperature is decreased when $\beta$ increases in order to maintain the same level of sharpness in the predictions no matter the value of $\beta$.
> Thus, an hypothesis that could connect to Veličković et al. (2024) is the fact optimization processes "prefers" a given level of sharpness. Also, since the level of sharpness changes with the batch size, using Sigmoids instead, as Zhai et al. (2023) suggests, would make the temperature to be invariant to changes in $\beta$. Intuitively, since the rest of the similarities are ignored for the Sigmoid, the fact that representations of negative pairs are closer in the space, should not affect the predictions.
> This point is pure speculation and excessive importance should not be attributed to it.
>
> > I think Equation 18 should be derived in the appendix.
>
> The derivation of this equation will be included in an appendix.
>
> > Table 2 needs to show std by running repeated experiments with different seeds.
>
> Please see the second answer to Reviewer DSZo.
>
> > I think preliminaries can be cut down to related work without the equations.
>
> We agree that the expressions of HSIC and CKA are not necessary for the understanding of the paper. They will be removed from the paper. Thanks for noticing.
>
> > Also, it would be great if you annotated the essence and nuisances in the examples.
>
> Essence and nuisances will be annotated in the examples. We suppose that you refer to the examples in section 3. Please let us know if you refer to other examples.
>
> We hope that your questions have been addressed. If this is the case and you consider that our paper deserves an increase in the score, we would thank you if you made it effective. If this is not the case, we are open to continue with the discussion in the next stage of the rebuttal process. Similarly, if you could elaborate on the question that we were unable to answer, we will try to address it.

---

> > ### Comment · Reviewer_Xy8R · 2025-04-07
> >
> > Thank you for addressing my concerns and questions. I have decided to increase the score to Accept.

---

### Official Review · Reviewer_ug8B · 2025-03-14

**Overall Recommendation:** 2

**Summary:**

The paper analyzes the problem that the contrastive losses in multimodal representation learning fail to align representations effectively due to their retention of modality-specific information.  To address this, the authors propose a variationally-derived regularization term that reduces modality-specific information, enhancing alignment based on the Information Bottleneck Principle.

**Claims And Evidence:**

The spherical Gaussian assumption for representation and indendical covariance for different modalities' representation are not supported by convincing evidence.  Such assumptions should be critically assessed and justified clearly in the context of general multimodal representation learning.

**Essential References Not Discussed:**

The ideas of essence and nuisances are similar to the unique and shared information concepts in [1]. The discussion between relevant references is missing.

[1]. Liang, Paul Pu, et al. "Factorized contrastive learning: Going beyond multi-view redundancy." Advances in Neural Information Processing Systems 36 (2023): 32971-32998.

**Experimental Designs Or Analyses:**

The soundness of experimental results on real-world tasks is weak.

1. Quantitative results of image retrievals are missing. It is not clear how well the model performs on this task.

2. The analysis of the toy example is interesting. However, providing the quantification of essence and nuisances and real-world dataset (e.g., image-text) will strengthen the soundness of the paper.

**Methods And Evaluation Criteria:**

Yes.

**Other Comments Or Suggestions:**

See my weaknesses and questions. If they are all addressed, I am willing to raise my score.

**Other Strengths And Weaknesses:**

It is interesting to see the analysis on the toy examples. However, I have the following concerns:

Is the spherical Gaussian assumption reasonable for general multimodal representation learning?
If the data are non-Gaussian, this assumption might be questionable. Particularly, considering your use of layer-wise normalization (e.g., with per-sample operations like those in the provided setting), it is difficult to justify that the learned representations strictly follow a simple Gaussian distribution.

Moreover, Eq. (20) assumes identical covariances for the two modalities, which is even less realistic given typical differences in modality-specific representation distributions. A more general and rigorous form that relaxes this assumption would strengthen the analysis.

**Questions For Authors:**

In Theorem 1, could you clarify what exactly $p(t|z_{\alpha})$ represents? The notation $t$ is not explained.

In Table 2, 0.1$L_M$ has the best CIDEr and BLEU scores. What will the model perform when having a stronger $L_M$ constraint? Do you have an ablation study on this?

**Relation To Broader Scientific Literature:**

This paper is relelated to the theoretical analysis on multimodal representation learning.

**Theoretical Claims:**

Yes.

---

> ### Author Rebuttal · Authors · 2025-03-31
>
> We would like to begin by thanking you for the time dedicated to give us feedback to improve our work. We address your main concerns next:
>
> > 1. Quantitative results of image retrievals are missing.
>
> > What will the model perform when having a stronger $L_M$ constraint?
>
> Please see second answer to Reviewer DSZo.
>
> > 2. providing the quantification of essence and nuisances and real-world dataset will strengthen the soundness of the paper.
>
> Quantifying the essence and nuisances is impossible in general because they are abstract variables that we simply define for our formulation. For this reason (and also because computing mutual information is expensive), it is in general impossible to quantify the essence and nuisances in the representation. In fact, this is the reason why we use variational approximations for the derivation of our loss function. If it were possible to exactly calculate the amount of the essence or nuisances in the representations, then we could simply maximize and minimize these quantities, respectively. The toy datasets are precisely used to set by ourselves the essence and nuisances and having a scenario in which it is straightforward to calculate a reliable estimation of the mutual information between the essence (or the nuisances) and the representations.
>
> > The ideas of essence and nuisances are similar to the unique and shared information concepts in [1].
>
> Thank you for your recommendation. This paper will be included in the related work section.
>
> > Is the spherical Gaussian assumption reasonable for general multimodal representation learning?
>
> On the one hand, the use of Gaussian distributions in the representation space is not an assumption but a design choice that we make mainly for tractability reasons of the KL divergence in equation 17. We argue its reasonability next.
>
> > If the data are non-Gaussian, this assumption might be questionable.
>
> On the other hand, the fact that the data distribution $p(x)$ is not Gaussian should not be problematic. For example, most of the VAEs (or other SoTA generative models, such as Diffusion Models and Flow Matching) use a encoder $p_\theta(z|x)$ that is Gaussian (also mainly for tractability reasons of the KL divergence) and they provide impressive performance in real-world applications, in which the data $p(x)$ is far from being Gaussian.
>
> > ...it is difficult to justify that the learned representations strictly follow a simple Gaussian distribution.
>
> Finally and most importantly, we must note that choosing the encoder $p_\theta(z|x)$  as Gaussian, does not imply at all that the representation space is Gaussian. What is Gaussian is the distribution of an embedding given a single input $p_\theta(z|x)$, but not the whole representation distribution $p_\theta(z)$.
>
> More specifically, we have that $p_\theta(z)=\int p_\theta(z|x)p(x)dx$. If we approximate the data distribution $p(x)$ as its empirical distribution given by $N$ datapoints, i.e., $p(x) \approx \sum_{i=1}^N \delta\left(x-x^{(i)}\right)$, then the distributions of the representation space $p_\theta(z)$ becomes a Gaussian mixture, i.e.,  $p_\theta(z) \approx \frac{1}{N} \sum_i \mathcal{N} \left(z; \mu_\theta\left(x^{(i)}\right), \sigma^2 I\right)$, which "is a universal approximator of densities, in the sense that any smooth density can be approximated with any speciﬁc nonzero amount of error by a Gaussian mixture model with enough components" [1].
>
> Furthermore, we note that using a stochastic encoder (even if it is Gaussian) results in a distribution that is richer than in the case of deterministic encoders (i.e., vanilla encoders). In the latter, $p_\theta(z|x)$ is a delta distribution, i.e., simpler (one parameter) than a gaussian (two parameters) and using deterministic encoders is rarely seen as problematic.
>
> [1] Goodfellow, I., Bengio, Y., & Courville, A. (2016). Deep learning., p. 65
>
> > Eq. (20) assumes identical covariances for the two modalities, which is even less realistic given typical differences in modality-specific representation distributions.
>
> Following with the previous answer, this should not be problematic. In fact, having the same covariance matrices in the representation spaces of both modalities could be considered as desirable since we would like these spaces to be as similar (or aligned) as possible.
>
> > The notation $t$ is not explained.
>
> $t$ means a realization of the task $T$. In probability theory, random variables are usually denoted by uppercase letters while their realizations are denoted by their corresponding lowercase letters (see https://en.wikipedia.org/wiki/Notation_in_probability_and_statistics). We will add a paragraph in section 2 to clarify this.
>
> We hope that your questions have been addressed. If this is the case and you consider that our paper deserves an increase in the score, we would thank you if you made it effective. If this is not the case, we are open to continue with the discussion in the next stage of the rebuttal process.

---

> > ### Comment · Reviewer_ug8B · 2025-04-06
> >
> > Thank the author for the response to my questions. However, some of my concerns still remain.
> >
> > I'm not assuming your $p(x)$ is Gaussian, but the latent representation $p(z)$ or $p(z|x)$. VAEs use a variational encoder to effectively parameterize a Gaussian latent space, which is reasonable.
> >
> > However, I'm confused now. Do you mean you are using a deterministic encoder to parameterize the latent distribution? How can this be a probabilistic approach, like VAEs, or capturing uncertainty? As you mentioned VAEs use Gaussian prior for the tractability reasons of the KL divergence. However, in your case, the KL divergence term becomes problematic when comparing a delta distribution with a standard Gaussian prior. In this sense, you are using $p(z)$ instead of $p(z|x)$ as the mapping is deterministic. If you use $p(z|x)$, why not calculate KL divergence instead of the L2 distance?
> >
> > Eq. 18 merely relies on the mean of the latent representation. It seems you calculate the mean of a batch of latent representations from two modalities and compare the L2 distance between them. In this sense, you are assuming $p(z)$ is Gassuian, which is questionable.
> >
> > Further, if Eq. 18 only measures the L2 distance between the mean of two modalities, InfoNCE has the same functionality.
> > According to [1], InfoNCE essentially contains the alignment term (see Sec. 4.1.1), which performs pairwise alignment (L2 distance). In this sense, the L2 distance between the mean is naturally minimized. The proposed loss is an enhancement on the alignment term with the same formulation.
> >
> > [1]. Wang T, Isola P. Understanding contrastive representation learning through alignment and uniformity on the hypersphere[C]//International conference on machine learning. PMLR, 2020: 9929-9939.
> >
> > " having the same covariance matrices in the representation spaces of both modalities could be considered as desirable since we would like these spaces to be as similar (or aligned) as possible.". This statement is not correct to me. Yes, we want two modalities to have the same covariance after alignment. However, if you cannot effectively parameterize their original distribution, like covariance, how can you optimize the two into the same covariance? Hence, many KDE-based methods, like MMD, require a kernel to effectively parameterize the shape of the distributions, which means simply assuming the two distributions have identical variance, e.g. 1, is not reasonable.

---

> > > ### Author Response · Authors · 2025-04-07
> > >
> > > Thank you for your answer. We believe some of our comments have been misunderstood and some concepts of variational inference have been mixed here. We try to clarify these points next.
> > >
> > > > I'm not assuming your $p(x)$ is Gaussian
> > >
> > > Of course, $p(x)$ **is not Gaussian in general** and, thus, this is never assumed.
> > >
> > > > but the latent representation $p(z)$ or $p(z|x)$
> > >
> > > The distribution $p(z|x)$  is chosen to be Gaussian in page 5. Then, as explained in our previous answer, we can marginalize as $p(z) = \int p(z|x)p(x)dx$. Thus, $p(z|x)$ **is Gaussian** but $p(z)$ **is not Gaussian**.
> > >
> > > > VAEs use a variational encoder to effectively parameterize a Gaussian latent space, which is reasonable.
> > >
> > > In VAEs, $p(z|x)$ is chosen to be Gaussian and $p(z)$ could be calculated by marginalizing, **exactly as in our case**.
> > >
> > > > Do you mean you are using a deterministic encoder to parameterize the latent distribution?
> > >
> > > If by *deterministic encoder* you mean that its parameters $\theta$ are deterministic, then our encoder is deterministic in the sense that it is not a Bayesian Neural Network.
> > >
> > > If by *deterministic encoder* you mean that it is used to model a delta distribution (i.e., it outputs a single embedding instead of a distribution of embeddings), then our encoder is not deterministic, since it outputs $p(z|x)$, i.e., a Gaussian distribution.
> > >
> > > To make it clear, **our encoder is exactly the same way as in VAEs** and it serves to obtain the parameters of a Gaussian distribution.
> > >
> > > > In this sense, you are using $p(z)$ instead of $p(z|x)$ as the mapping is deterministic.
> > >
> > > No, not at all, $p(z|x)$ is the distribution that we are considering to optimize equation 18. The process to calculate equation 18 is very simple:
> > > 1. Given a batch of pairs of inputs of modalities $\alpha$ and $\beta$, i.e., $\left\lbrace x_\alpha^{(i)}, x_\beta^{(i)} \right\rbrace\_{i=1}^N$, we calculate the output distributions $p\left(z|x_\alpha^{(i)}\right)$ and $p\left(z|x_\beta^{(i)}\right)$ for $i=1,\dots,N$. We choose $p\left(z|x_\alpha^{(i)}\right)$ and $p\left(z|x_\beta^{(i)}\right)$ to be Gaussians.
> > > 2. We calculate the KL divergence between each of pair of distributions, i.e., $D\_{KL} \left( p\left(z|x_\alpha^{(i)}\right) \mid \mid p\left(z|x_\beta^{(i)}\right) \right)$ for $i=1,2,\dots,N$. Since these distributions are Gaussians, this term is tractable.
> > > 3. We estimate the expectation of the KL Divergences as $\frac{1}{N}\sum\_{i=1}^N D\_{KL} \left( p\left(z|x_\alpha^{(i)}\right) \mid \mid p\left(z|x_\beta^{(i)}\right) \right)$.
> > >
> > > > If you use $p(z|x)$, why not calculate KL divergence instead of the L2 distance?
> > >
> > > We are minimizing the KL divergence, which is equivalent to minimizing the L2 distance when the covariance matrices are constant (see https://mr-easy.github.io/2020-04-16-kl-divergence-between-2-gaussian-distributions/).
> > >
> > > > Eq. 18 merely relies on the mean of the latent representation.
> > >
> > > It relies on the mean of $p(z|x)$ but not on the mean of $p(z)$. As it is clearly stated in equation 18, the KL divergence of $p(z|x)$ is calculated first and the expectation of this is calculated afterwards.
> > >
> > > > It seems you calculate the mean of a batch of latent representations from two modalities and compare the L2 distance between them.
> > >
> > > No, not at all. We do not do this and we do not know where it seems that. As stated in the previous answer, we calculate the expectation of the KL divergences and not the KL divergences of the expectation.
> > >
> > > > In this sense, you are assuming $p(z)$ is Gassuian, which is questionable.
> > >
> > > As previously stated, **$p(z)$ is never assumed to be Gaussian**. What is Gaussian is $p(z|x)$, but not $p(z)$.
> > >
> > > > According to [1], InfoNCE essentially contains the alignment term (see Sec. 4.1.1), which performs pairwise alignment.
> > >
> > > As demonstrated in Theorem 4.1 of [1], InfoNCE global minimum happens when a perfect alignment exists **but when the number of negative samples tends to infinite**. One of the main contribution of our work is Theorem 2, which demonstrates that misalignment is caused by information misbalance. Then, the solution for the information misbalance when the encoders are Gaussian results in the alignment definition given in [1], thus making our proposal consistent with the literature. But we note, that we give a theoretical derivation which is more general than the one in [1], since the latter is a particular case of ours.
> > >
> > > > many KDE-based methods, like MMD, require a kernel to effectively parameterize the shape of the distributions...
> > >
> > > Density estimation and what we are doing are two different worlds. In our case, **$z$ is defined through $p(z|x)$, so this is the original distribution**. In KDE methods, it is assumed that an unkown true distribution generated a set of data, and some methods are used to find distributions that could have likely generated the given set of data. In our case, it is ourselves who are defining $z$, so we know its true distribution, which is $p(z|x)$.

---

### Official Review · Reviewer_DSZo · 2025-03-18

**Overall Recommendation:** 3

**Summary:**

This paper addresses the challenge of misalignment in multimodal representation learning when using contrastive loss functions. The authors argue that this misalignment stems from modality-specific information present in the representation space that contrastive objectives fail to remove. Leveraging the Information Bottleneck Principle, they provide a theoretical framework to explain this phenomenon and propose a novel regularization term that enhances representational alignment by reducing modality specific information and find the sufficient minimum information. Through empirical validation, the authors demonstrate that their approach not only improves alignment but also enhances performance in real-world tasks such as image captioning, aiming at highlighting that balancing information preservation and compression in multimodal learning is important.

**Claims And Evidence:**

The claims in the paper are generally well-supported by both theoretical and empirical studies, with the authors providing meaningful contributions in both areas. The theoretical framework leverages the Information Bottleneck Principle to explain misalignment caused by modality-specific nuisances, while empirical validation includes controlled experiments and real-world applications (e.g., image captioning), demonstrating the effectiveness of their proposed regularization term.

However, certain claims require further substantiation. The paper assumes that modality-specific information always contributes to misalignment, but this may depend on the dataset and task. More evidence is needed to establish the universality of this claim. Additionally, the Information Homeostasis phenomenon—where encoders purportedly adjust internal parameters to preserve nuisance information—is an intriguing hypothesis but lacks deeper causal analysis. Conducting additional experiments to isolate confounding factors would strengthen this argument.

Moreover, the paper does not compare its method against several state-of-the-art techniques designed for alignment beyond contrastive learning. Including such comparisons would provide a clearer assessment of its advantages and limitations.

**Essential References Not Discussed:**

Some key works in the area that could be added: adversarial alignment (Lample et al., NeurIPS 2018), cross-modal transformers (imagebind of Girdhar et al., CVPR 2023), minimum sufficient representation learning (Wang et al., CVPR 2022), platonic Representation Hypothesis (Huh et al., 2024 - https://arxiv.org/abs/2405.07987) for self-supervised, multimodal IB (https://arxiv.org/abs/2210.17444), PID appraoches (https://arxiv.org/pdf/2401.13503, https://arxiv.org/pdf/2409.07402v1 (for alignement), and https://arxiv.org/pdf/2402.06223v1 (Multimodal Contrastive Representation Learning through Latent Partial Causal Models).

**Experimental Designs Or Analyses:**

The experimental design in the paper is generally sound and well-structured, combining controlled experiments on disentanglement datasets with real-world applications. The controlled experiments effectively analyze the impact of different factors on alignment and nuisance removal, while the image captioning task serves as a strong practical validation. The use of CKA as a metric is appropriate, but incorporating additional task-specific evaluations, such as retrieval accuracy, could further strengthen the analysis. A key limitation is the lack of comparisons with state-of-the-art alignment methods beyond contrastive learning, which would provide a clearer perspective on the proposed method’s relative advantages and potential shortcomings. Additionally, the study of the information homeostasis phenomenon, while interesting and relevant, lacks causal analysis, making it difficult to fully substantiate its claims. Conducting more ablation studies to isolate potential confounding factors would improve the robustness of this finding.

**Methods And Evaluation Criteria:**

The proposed methods and evaluation criteria are generally speaking well-suited to the problem of multimodal representation alignment. The authors leverage the IB principle to design a regularization term that explicitly reduces modality-specific nuisances, effectively addressing misalignment. Their evaluation includes both controlled experiments and real-world applications, providing a solid empirical foundation.

However, there is room for improvement:
(i) Expanding the evaluation to include additional multimodal benchmarks would enhance generalizability.
(ii) Incorporating additional task-specific metrics beyond alignment (e.g., retrieval accuracy, downstream task performance) would strengthen the assessment.
(iii) Rather than solely validating the proposed approach, benchmarking against existing alignment techniques (e.g., cross-modal transformers, adversarial methods) would provide a clearer comparison and better contextualize the contribution of this work.

**Other Comments Or Suggestions:**

Most questions have been raised in the previous sections but a key one is how this approach is related to partial information decomposition (PID). PID seems to be relevant to this problem and can be leveraged to formalize some of the claims.

**Other Strengths And Weaknesses:**

The strengths and the weaknesses have been discussed in the previous sections. Nothing major to add here.

**Questions For Authors:**

N/A

**Relation To Broader Scientific Literature:**

The key contributions of the paper build upon and extend foundational ideas in multimodal learning, contrastive learning, and information theory. It is closely related to prior work on contrastive representation learning (Oord et al., 2018; Radford et al., 2021), which aims to align representations by maximizing mutual information between different modalities. However, contrastive objectives alone have been shown to be insufficient for achieving true alignment due to the presence of modality-specific nuisances (Liang et al., 2022). This paper addresses this limitation by leveraging the Information Bottleneck Principle (Tishby et al., 2000; Alemi et al., 2016; Achille & Soatto, 2018) to derive a principled regularization term that explicitly reduces nuisances, providing a more structured approach than heuristic modifications proposed in prior works (Li et al., 2021; 2022; 2023). The study also aligns with recent efforts in multi-view learning (Tian et al., 2020) and information-based disentanglement (Wang et al., 2022), which emphasize the role of minimal sufficient representations in improving alignment. Additionally, the paper introduces the Information Homeostasis hypothesis, which suggests that encoders adjust internal parameters to maintain a balance in information entropy—a concept related to implicit regularization in deep networks (Shwartz-Ziv & Tishby, 2017) but not yet extensively studied in the context of multimodal alignment. Nevertheless, incorporating additional literature and direct comparisons with alternative alignment techniques would further strengthen its positioning within the broader research landscape.

**Theoretical Claims:**

The paper presents several theoretical claims, primarily leveraging the IB principle to explain modality-specific misalignment. The proofs for key results, including Theorems 1, 2, and Lemma 1, appear correct and logically sound. However, some aspects could be further clarified or refined. For instance, a formal guarantee on how different encoders may lead to equivalent partitions in practical settings could be added in Lemma 1. Theorem 1 assumes that tasks derived from the essence fully capture all relevant downstream tasks, which may not always hold universally across applications. Theorem 2 relies on the assumption that perfect alignment can only be achieved if modality-specific information is entirely removed, which may need fine-tuning, as reasonable alignment can still be attained in practice even if some nuisances remain. While the theoretical foundations are solid, addressing these points would enhance the rigor and practical relevance of the claims.

---

> ### Author Rebuttal · Authors · 2025-03-31
>
> We would like to begin by thanking you for the time dedicated to give us feedback to improve our work. We address your main concerns next:
>
> > (i) additional multimodal benchmarks would enhance generalizability.
>
> More experiments were not included due to space limitations.
>
> > (ii) task-specific metrics beyond alignment would strengthen the assessment.
>
> In section 5, we have Fig.6 for this purpose. For section 6, we have calculated retrievals, shown next.
>
> ||CIDEr|BLEU@4|I2T R@1|T2I R@1|
> |-|-|-|-|-|
> |ITC+LM|$91.7\pm 0.2$|$28.6\pm 0.1$|$64.2\pm 0.2$|$52.3\pm 0.4$|
> |ITC+LM+ITM|$91.8\pm 0.5$|$28.8\pm 0.2$|$61.4\pm 0.6$|$49.7\pm 0.8$|
> |ITC+LM+$0.01\mathcal{L}_M$|$92.3\pm 0.8$|$29.1\pm 0.4$|$64.0\pm 0.3$|$52.3\pm 0.5$|
> |ITC+LM+$0.03\mathcal{L}_M$|$92.6\pm 0.3$|$29.2\pm 0.2$|$63.9\pm 0.4$|$52.1\pm 0.5$|
> |ITC+LM+$0.1\mathcal{L}_M$|$93.0\pm 0.3$|$29.4\pm 0.3$|$63.0\pm 0.5$|$50.4\pm 0.5$|
> |ITC+LM+$0.3\mathcal{L}_M$|$90.5\pm 0.4$|$28.5\pm 0.2$|$59.6\pm 0.4$|$47.1\pm 0.4$|
>
> We have that:
> 1. Text generation (TG) and retrieval performances are inversely correlated. This makes sense, since TG gets benefited from minimal representations and retrieval from sufficient representations (retrieval is a specific type of downstream task). This inverse correlation can be observed also in Fig. 6.
> 2. Our loss increases TG performance for low or medium values of $\beta$, since its goal is to increase the minimality of representations.
> 3. For $\beta=0.3$, the performance starts to excessively decrease the part of the essence that representations retain, similarly to in Fig. 6.
>
> This would be included and better explained in the final version.
>
> > additional experiments to isolate confounding factors would strengthen this argument.
>
> We agree that the experimental setup that analyzes the Information Homeostasis phenomenon does not serve to demonstrate any causal relationship. However, this analysis would require more space and, additionally, it could confuse the main line of the paper.
>
> > the paper does not compare its method against several state-of-the-art techniques designed for alignment...
>
> We compare in section 6 our loss function with the ITM, which is used in most of the state-of-the-art methods to obtain alignment between text and image modalities. As explained in lines 379-382, ITM is defined for a very specific architectural choice so it cannot be used in experiments in section 5.
>
> > a formal guarantee on how different encoders may lead to equivalent partitions in practical settings could be added in Lemma 1.
>
> We are not sure to understand this point very clearly. Lemma 1 is stated in line 134 and representations are defined in line 161, so there is no notion of encoder or representation at the point in which Lemma 1 is stated. Thus, this lemma is completely independent of the encoders.
>
> > Theorem 1 assumes that tasks derived from the essence fully capture all relevant downstream tasks, which may not always hold universally across applications.
>
> This is never assumed. Theorem 1 simply states that sufficient representations are a necessary condition to solve all the tasks that can be derived from the essence. Sometimes, contrastively trained models are used to solve downstream tasks that are not in the essence. However, there are no theoretical guarantees that this should work no matter if the representation is sufficient or not.
>
> > The paper assumes that modality-specific information always contributes to misalignmentt, but this may depend on the dataset and task.
>
> > Theorem 2 relies on the assumption that perfect alignment can only be achieved if modality-specific information is entirely removed.
>
> This is not an assumption but a theorem that is demonstrated in Appendix A.3. If the demonstration is correct, then it is universally true. Apart from that, alignment is totally independent of the task, since it is an intrinsic property of the representations. We are open to elaborate on this, but we are not sure to understand your point.
>
> > Essential References Not Discussed
>
> Thank you for the given references. Some of them were already mentioned in the paper and the rest of them will be discussed in the related work.
>
> > how this approach is related to PID.
>
> We believe that the connection is very weak. PID analyzes how two source inputs contribute to the information in a target variable. This is relevant in the cases in which we are using two inputs at the same time, but this is rarely the case in multimodal learning. The main connection that we could make between our work and PID is that, in case that we wanted to use both representations to solve a downstream task, then the redundant information would be equal to zero for minimal representations.
>
> We hope that your questions have been addressed. If this is the case and you consider that our paper deserves an increase in the score, we would thank you if you made it effective. If this is not the case, we are open to continue with the discussion in the next stage of the rebuttal.

---

> > ### Comment · Reviewer_DSZo · 2025-04-09
> >
> > Thank you for addressing my concerns and questions and for providing clarifications. Thank you as well for the proposed additions in the final version (e.g., second comment).
> >
> > Nevertheless, some clarifications are necessary to avoid a potential misunderstanding.
> >
> > Regarding the questions on the theoretical results (mainly Lemma 1, and Theorems 1-2).
> > Indeed, Lemma 1 is formulated entirely in information-theoretic terms, before introducing encoders or representations, and is thus independent of any practical implementation. However, given that the theoretical results are intended to support the paper's overall objective and empirical evaluation, and that the venue focuses on AI/ML rather than purely theoretical information theory, a discussion of their practical relevance and applicability would strengthen the contribution (and is expected in a comprehensive study).  So, while the lemma is valid under idealized assumptions, one can expect in real-world systems, due to various impairments, two encoders even trained on the same datasets may not induce representations that are bijectively related, and thus may fail to yield equivalent partitions of the input space, even if they aim to capture the same essence. We tend to believe that there is a gap between theory and practice, which could deserve acknowledgment or further discussion/study.
> >
> > A similar perspective applies to Theorem 2. The concern is not related to the correctness of the theoretical result, but mainly to the proof’s assumptions and the result’s applicability in real-world systems. The theorem implies that perfect alignment is achieved only when all modality-specific information (nuisances) is removed. However, learned representations typically retain some modality-specific content (isn’t this the case in section 5, e.g., Fig. 5?). Therefore, if perfect alignment requires or implies complete removal or absence of nuisances, a discussion on the practical feasibility of this condition would be beneficial. It could be that partial minimization of nuisances is a sufficient and meaningful proxy and serves as an effective approximation, but this has to be shown. Finally, regarding the statement that alignment is "totally independent of the task", it would be helpful to clarify whether this refers to the geometric property of the learned representation space rather than downstream performance. This would improve clarity and prevent misinterpretation. We hope the original review comments are clarified.
> >
> > Finally, regarding the potential application of PID to multimodal scenarios, see: https://arxiv.org/html/2302.12247v5 although the links might be deeper and rooted in the information-theoretic formulation (https://arxiv.org/html/2405.07665v1).

---

### Decision · Program_Chairs · 2025-05-01

**Decision:**

Accept (poster)

**Comment:**

This paper has overall positive scores, except some concerns from reviewer ug8B. These concerns seem to have been addressed in a adequate manner by the authors and the concerns were not shared by the other reviewers. The other reviewers (in particular reviewers DSZo and N9GW) highlight that the theory and experimental design are overall sound. The theoretical part and the methodology bring a useful contribution to the field.